# EFFICIENT FEDERATED DOMAIN TRANSLATION

**Zeyu Zhou, Sheikh Shams Azam,** **Christopher Brinton, David I. Inouye**
Elmore Family School of ECE, Purdue University
`{zhou1059, azam1, cgb, dinouye}@purdue.edu`

## ABSTRACT

A central theme in federated learning (FL) is the fact that client data distributions are often not independent and identically distributed (IID), which has strong implications on the training process. While most existing FL algorithms focus on the conventional non-IID setting of class imbalance or missing classes across clients, in practice, the distribution differences could be more complex, e.g., changes in class conditional (domain) distributions. In this paper, we consider this complex case in FL wherein each client has access to only one domain distribution. For tasks such as domain generalization, most existing learning algorithms require access to data from multiple clients (i.e., from multiple domains) during training, which is prohibitive in FL. To address this challenge, we propose a federated domain translation method that generates pseudodata for each client which could be useful for multiple downstream learning tasks. We empirically demonstrate that our translation model is more resource-efficient (in terms of both communication and computation) and easier to train in an FL setting than standard domain translation methods. Furthermore, we demonstrate that the learned translation model enables use of state-of-the-art domain generalization methods in a federated setting, which enhances accuracy and robustness to increases in the synchronization period compared to existing methodology.

## 1 INTRODUCTION

Distribution shift across clients is a well-known challenge in the Federated Learning (FL) community (Huang et al., 2021). Most existing works have considered this from the perspective of class imbalance or missing classes (i.e., a shift in marginal distribution of classes) across clients, a form of non independent and identically distributed (IID) datasets (Zhao et al., 2018). In particular, these works typically assume implicitly that the class conditional distribution of data is the same. In practice, however, the conditional distributions across different clients could be very different, e.g., in computer vision, there is a shift in the data distribution (specifically, illumination) of images captured during the day versus night irrespective of the class label (Lengyel et al., 2021). This can lead to significant model generalization errors even if we solve the issue of class shifts. Translating between datasets is one promising strategy for mitigating the more general shift across distributions of different clients. Moreover, it could solve the problem of Domain Generalization (DG) which requires a model to generalize to unseen domains (Nguyen et al., 2021).

A domain translation model is one that can translate data from different distributions, typically attempting to align the conditional shift across distributions. In centralized settings, many translation methods have been proposed, such as StarGAN (Choi et al., 2018). However, in FL, domain translation models can be difficult to train because most existing methods require access to data across all domains. Prior literature does not consider this natural setting of *federated domain translation* where domain datasets are distributed across clients.

In this paper, we empirically demonstrate that a naive implementation of state-of-the-art (SOTA) translation models in the FL context indeed performs poorly given communication limitations between the server and clients that often exist in practice (Azam et al., 2022a). Then, we propose leveraging an iterative translation model, Iterative Naive Barycenter (INB) (Zhou et al., 2022), which is much more amenable to FL training in terms of communication efficiency and data privacy considerations. We

---

*Currently at Apple.

empirically demonstrate that this modification obtains far superior performance to standard translation methods in the FL setting, and that it can aid in solving the challenge of DG in FL settings.

Our main contributions are summarized as follows:

- We develop a federated domain translation methodology based on the recent iterative approach INB, which is more amenable to the FL setting than standard translation methods. We analytically show the equivalence between our federated algorithm and original INB which is important for enabling usage of INB in the federated setting.

- We further propose several FL-motivated improvements to INB, including the use of variable-bin-width histograms, which significantly reduce communication costs.

- We empirically demonstrate that our FedINB approach performs significantly better than standard translation models under the practical limited communication setting.

- As one application, we demonstrate the feasibility of leveraging our federated translation model to aid in federated domain generalization. We also show that our federated DG method provides substantial improvements in robustness to an increasing synchronization period, allowing reductions in communication overhead.

## 1.1 BACKGROUND: UNPAIRED TRANSLATION METHODS

Unpaired domain translation is the task of learning to translate between every pair of domains using only unpaired samples from each domain (Zhu et al., 2017). Formally, let $M$ be the number of domains and $p_m(x)$ denote the true $m$-th domain distribution. Let $\mathcal{X}_m = \{x_m^{(i)} \sim p_m\}_{i=1}^{n_m}$ denote the training dataset from the $m$-th domain distribution, where $x_m^{(i)} \in \mathbb{R}^d$, $n$ is the number of samples per domain, and $d$ is the number of dimensions. Also, let $f_{m \to m'}$ denote the translation model from the $m$-th domain to the $m'$-th domain. Given this notation, the translation problem is usually formulated as minimizing a distribution divergence $D$ (e.g., Jensen-Shannon Distance (JSD) for adversarial learning) between the translated and true distributions with some regularization term $R$:

$$\min_{\{f_{m \to m'}\}_{m \neq m'}} \sum_{m=1}^{M} \sum_{m' \neq m} D(\hat{p}_{f_{m \to m'}}, p_{m'}) + \lambda R(f_{m \to m'}) \tag{1}$$

where $\hat{p}_{f_{m \to m'}}$ is the distribution of the samples translated from the $m$-th domain to the $m'$-th domain, i.e., the distribution of $f_{m \to m'}(x_m)$ where $x_m \sim p_m$.

**Standard GAN-based Translation Methods**  Zhu et al. (2017) proposes CycleGAN, which estimates unpaired translation models between two domains, using adversarial loss to approximate the divergence term and cycle consistency loss for the regularization term. StarGAN (Choi et al., 2018) extends CycleGAN by proposing a unified model for domain translation between multiple domains using a single translation model that takes the source and target domain labels as input. A key issue with most existing translation models is that the computation of their objective requires access to data from all domains in the training, which is prohibited in an FL setting. For example, in StarGAN, to compute the domain classification loss for fake data, we need a discriminator trained on other domains. While the issue could be mitigated by federated algorithms such as FedAvg (McMahan et al., 2017), this requires frequent global synchronization across domains and can be hard to train as we show in Section 4. While more advanced unified translation models exist (e.g., StarGANv2 (Choi et al., 2020)), they are trained in similar ways to StarGAN and will suffer from the same drawbacks. Besides, many existing translation models learn pairwise translation (Zhu et al., 2017; Park et al., 2020) which would require an excessive computation and communication effort as the number of clients in an FL setting increases. Thus, we focus on StarGAN in our experiments as an archetype model of standard translation methods.

**Iterative Naive Barycenter (INB)**  In contrast to standard translation approaches, the Iterative Naive Barycenter (INB) method (Zhou et al., 2022) builds up a deep translation model by solving a sequence of much simpler problems that are highly amenable to the FL setting (as will be described in the next section). INB learns deep invertible transformations $T_m = t_m^{(L)} \circ \cdots \circ t_m^{(1)}$ (where $L$ is the number of layers) that map each domain distribution to a shared latent distribution. Given these invertible

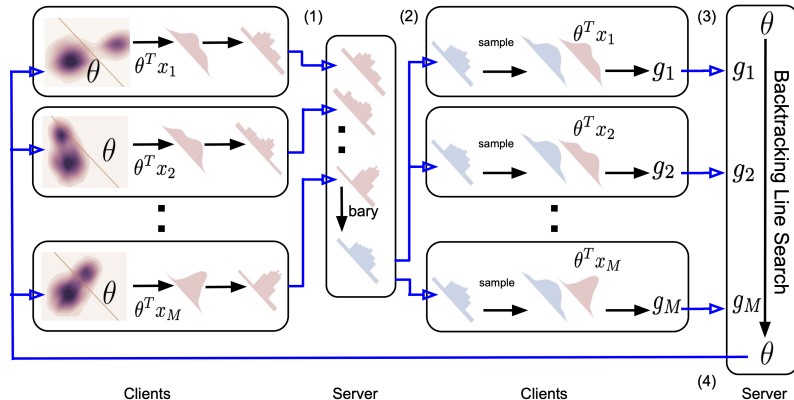

Figure 1: High-level visualization of our Fed-multi-max-K-SW methodology with VW histograms. At each iteration, there are 4 transmissions: (1) histograms of data from each domain, (2) histograms of the empirical barycenter, (3) gradients $\boldsymbol{g}_m$, and (4) updated projection $\boldsymbol{\theta}$. These transmissions are the main sources of communication cost of FedINB which we minimize in this paper.

transformations, a domain translation model can then be expressed as $f_{m \to m'} \triangleq T_{m'}^{-1} \circ T_m$, i.e., translate to the shared latent distribution and then inverse translate to any other domain. Importantly, each invertible layer $t^{(\ell)}$ is fit greedily by solving two simple problems: (1) finding the projection matrix that maximizes the multi-distribution K-sliced Wasserstein divergence (Zhou et al., 2022), and (2) solving 1D Wasserstein barycenter problems along the projection directions. Because 1D Wasserstein barycenter problems are known to have closed-form solutions in terms of the domain 1D CDFs, the key computational challenge is solving (1), which is formally defined as:

$$\max_{\boldsymbol{\theta}:\boldsymbol{\theta}^T\boldsymbol{\theta}=I_K} \frac{1}{MKn} \sum_{m=1}^{M} \sum_{k=1}^{K} \sum_{i=1}^{n} |(\boldsymbol{\theta}_k^T \boldsymbol{x}_m)_{[i]} - \boldsymbol{y}_{[i],k}|^2, \qquad (2)$$

where $\boldsymbol{x}_m \in \mathbb{R}^{d \times n}$ is the sample data matrix for the $m$-th domain, $\boldsymbol{\theta} = [\boldsymbol{\theta}_1, \dots, \boldsymbol{\theta}_K]$, $(\boldsymbol{\theta}_k^T \boldsymbol{x}_m)_{[i]}$ signify the samples from the $m$-th domain distribution projected along the direction $\boldsymbol{\theta}_k$ sorted in ascending order, $\boldsymbol{y}_{[i],k} \triangleq \frac{1}{M} \sum_{m=1}^{M} (\boldsymbol{\theta}_k^T \boldsymbol{x}_m)_{[i]}$ is the empirical barycenter along direction $\boldsymbol{\theta}_k$, $d$ is the dimension of the data, $K \leq d$ is the number of projection directions, $I_K \in \mathbb{R}^{K \times K}$ is the identity matrix, $i$ is a vector index. and $[i]$ is the sorted index. The corresponding algorithm is called multi-max-K-SW (Zhou et al., 2022, Algorithm 3). More details of INB can be found in Appendix C.

## 2 FEDERATED INB

### 2.1 PROBLEM STATEMENT: FEDERATED TRANSLATION WITH CLASS-CONDITIONAL SHIFTS

We will make a natural assumption that there is a *one-to-one mapping* between clients and domains, i.e., each client only has access to data from its own domain. We will also extend the unpaired translation task to consider both a data instance $x$ and its corresponding class label $y$, where $p_m(x, y)$ will denote the joint distribution of $x$ and $y$ for the $m$-th domain. For this extension, we seek a class-conditional translation model $f_{m \to m'}(x|y)$ which aims to translate from $p_m(x|y)$ to $p_{m'}(x|y)$ given a class label $y$. Ultimately, our goal is to learn domain translation models across all clients such that they can be applied for other downstream federated tasks. Most existing non-IID FL works assume that the client distributions only exhibit class imbalance, i.e., the marginal class distributions are different ($p_m(y) \neq p_{m'}(y)$), but the class-conditional distributions are equal ($p_m(x|y) = p_{m'}(x|y)$). In contrast, we focus on the case where the class-conditional distributions differ across clients.

### 2.2 FEDERATED MAXIMUM MULTI-DISTRIBUTION K-SLICED WASSERSTEIN DIVERGENCE

In Algorithm 1, we adapt the original multi-max-K-SW algorithm to show explicitly where computation is done and when communication is needed in FL. For example, {Server} means the computation is done on the central server and {Server $\to$ Clients} means the following data will need to be

transmitted from the server to all clients. A major problem with implementing INB in the federated setting is the loss of gradient informtation during transmission. In particular, the empirical barycenter $\boldsymbol{y}_{[i],k} \triangleq \frac{1}{M} \sum_{m=1}^{M} (\boldsymbol{\theta}_k^T \boldsymbol{x}_m)_{[i]}$ becomes a constant (because it is transmitted and the gradient is not tracked) instead of a function of $\boldsymbol{\theta}$. However, we show that treating $\boldsymbol{y}_{[i],k}$ as a constant will actually return the same gradient value. Formally, we make the following remark (proof in Appendix B).

**Remark 1.** Fed-multi-max-K-SW *and* multi-max-K-SW *compute equivalent results despite* Fed-multi-max-K-SW *treating* $\boldsymbol{y}_{[i],k}$ *as a constant rather than a function of* $\theta$.

*Proof sketch.* Each client computes a biased gradient $\nabla_{\boldsymbol{\theta}} \boldsymbol{d}_m = \nabla_{\boldsymbol{\theta}}^{\text{true}} \boldsymbol{d}_m + \boldsymbol{b}_m$ where $\boldsymbol{b}_m$ is the bias of the gradient estimate. However, for the special case where the cost function is $c(\boldsymbol{x}, \boldsymbol{y}) = \|\boldsymbol{x} - \boldsymbol{y}\|_2^2$, these biases cancel out each other so that the sum of the biased client gradients is equal to the true gradient, i.e., $\sum_m \nabla_{\boldsymbol{\theta}} \boldsymbol{d}_m = \sum_m \nabla_{\boldsymbol{\theta}}^{\text{true}} \boldsymbol{d}_m + \sum_m \boldsymbol{b}_m = \sum_m \nabla_{\boldsymbol{\theta}}^{\text{true}} \boldsymbol{d}_m$ because $\sum_m \boldsymbol{b}_m = 0$.

---

**Algorithm 1** Fed-multi-max-K-SW

---

**Input:** Samples from the $M$ class distributions $(\boldsymbol{x}_1, \boldsymbol{x}_2, \ldots, \boldsymbol{x}_M)$, number of directions $K$, max number of iterations $J$
**Output:** Estimated projection matrix $\boldsymbol{\theta}$
  {Server} Randomly initialize $\boldsymbol{\theta} \in \mathbb{R}^{d \times K}$ satisfying $\boldsymbol{\theta}^T \boldsymbol{\theta} = I_K, \boldsymbol{\theta} = [\boldsymbol{\theta}_1, \ldots, \boldsymbol{\theta}_K]$
  {Server $\to$ Clients} $\boldsymbol{\theta}$
  **for** $j = \{1, 2, \ldots, J\}$ **do**
    {Clients} Compute projections and sort samples along each 1D projection $(\boldsymbol{\theta}_k^T \boldsymbol{x}_m)_{[i]}$
    {Clients $\to$ Server} $(\boldsymbol{\theta}_k^T \boldsymbol{x}_m)_{[i]}$
    {Server} Compute empirical barycenter $\boldsymbol{y}_{[i],k} = \frac{1}{M} \sum_{m=1}^{M} (\boldsymbol{\theta}_k^T \boldsymbol{x}_m)_{[i]}$
    {Server $\to$ Clients} $\boldsymbol{y}_{[i],k}$
    {Clients} Compute local objective $\boldsymbol{d}_m = -\frac{1}{MKn} \sum_{k=1}^{K} \sum_{i=1}^{n} |(\boldsymbol{\theta}_k^T \boldsymbol{x}_m)_{[i]} - \boldsymbol{y}_{[i],k}|^2$
    {Clients} Compute gradient $\boldsymbol{g}_m = \nabla_{\boldsymbol{\theta}} \boldsymbol{d}_m$
    {Clients $\to$ Server} $\boldsymbol{g}_m$
    {Server} Aggregate gradients $\boldsymbol{g} = \sum_m \boldsymbol{g}_m, \boldsymbol{u} = [\boldsymbol{g}, \boldsymbol{\theta}], \boldsymbol{v} = [\boldsymbol{\theta}, -\boldsymbol{g}]$
    {Server} Update with backtracking line search $\boldsymbol{\theta} = \boldsymbol{\theta} - \tau \boldsymbol{u}(I_{2K} + \frac{\tau}{2} \boldsymbol{v}^T \boldsymbol{u})^{-1} \boldsymbol{v}^T \boldsymbol{\theta}$
    {Server $\to$ Clients} $\boldsymbol{\theta}$
    **if** $\boldsymbol{\theta}$ converge **then** Stop **end if**
  **end for**

---

## 2.3 FEDERATED 1D BARYCENTER

The next step of INB is to solve the 1D Wasserstein barycenter problems along each direction independently and estimate the corresponding invertible transportation map between marginal distribution and barycenter. The procedure can be found in Algorithm 2. We first estimate histogram densities at each client and send those histogram bins to the server. Then we compute the inverse CDF of the barycenter at the server and send the histogram bins of the inverse barycenter back to each client. We summarize the full algorithm of FedINB in Algorithm 3 of Appendix C.

## 2.4 VARIABLE-BIN-WIDTH HISTOGRAMS FOR MAX-K-SWD

Intuitively, the sorting operation on the clients in Algorithm 1 will improve data privacy because the samples are sorted independently for each projection direction. The joint dependencies between samples are broken and thus joint samples cannot be reconstructed from these sorted samples. From a distribution perspective, after sorting each direction *independently*, the transmitted data only contains information about the *marginals* of the domain distributions along the projection directions rather than information about the *joint* domain distributions. However, in practice, there may still be a concern that real data is being transmitted, as is often prohibited in FL. Additionally, since we are sending all data along those projection, the number of parameters being transmitted will increase linearly with the number of samples, which could lead to a high communication cost. Hence, we propose to use histogram density estimators to approximate the empirical slice Wasserstein distance objective in multi-max-K-SW. Specifically, instead of sending data directly, we first fit histogram

densities at each client and send the histogram bins to the server. To avoid high computational cost, we use Variable-Bin-Width histograms (VW histograms) as defined next.

**Definition 1** (Variable-Bin-Width histograms)*. Given a number of bins $B$, the probability of each bin is assumed to be equal to $1/B$ while the bin edges $b_i$ are selected as the uniformly spaced quantiles of the empirical distribution, i.e., $b_0 = \hat{F}^{-1}(0), b_i = \hat{F}^{-1}(i/B) \ \forall i \in \{1, \ldots, B\}$, where $\hat{F}^{-1}$ is a pseudo-inverse of the empirical CDF of the samples and $B$ is the number of bin edges.*

By this definition, the samples in each bin are the same (assuming the number of samples is exactly divisible by the number of bins). While a standard histogram has uniformly spaced bin edges where the probability of each bin depends on the data, VW histograms have uniform spaced cumulative probabilities where the bin edges depend on the data. A benefit of using VW histograms is that the computation of 1D barycenter histograms is merely the average of histogram bin edges because the barycenter is based on the inverse CDF (in contrast, a mixture model is based on mixing the CDF). After computing the barycenter histograms, we can readily employ the inverse CDF with uniformly spaced quantiles to estimate pseudo empirical barycenter samples $y'_{[i],k} \triangleq \hat{F}^{-1}_{k,\text{bary}}(i/n)$:

$$d_m = \frac{w_m}{K} \sum_{k=1}^K \frac{1}{n_m} \sum_{i=1}^{n_m} |(\boldsymbol{\theta}_k^T \boldsymbol{x}_m)_{[i]} - y'_{[i],k}|^2 \,. \tag{3}$$

The visualization of Algorithm 1 with VW histograms can be found in Figure 1. Beyond communication cost, VW histograms also provides improved data privacy as it sends a smoothed and compressed version of the marginal distributions rather than sending the sorted real samples. From an information theory perspective, VW histograms send less information because the histogram estimation is not a one-to-one mapping. A more comprehensive discussion on privacy can be found in Appendix D.

As the number of bins increases, the algorithm approaches the original Fed-multi-max-K-SW (more details in Appendix C.2). We empirically show that using limited number of bins is enough to lead to a good optimization result and may even be preferred over a large number of bins because it may regularize the problem. To summarize, using VW histograms in FedINB improves communication costs and privacy while also possibly improving translation performance due to its regularizing effect.

## 2.5 PRACTICAL IMPROVEMENTS

We briefly introduce two practical improvements that can reduce communication costs further. The first simplifies the line search procedure of the original multi-max-K-SW algorithm and is discussed in Appendix C.3. The second improvement uses autoencoders to reduce dimensionality because the computational and communication cost of the algorithm scale linearly with the data dimension $d$ (as detailed in Section 2.6). Also, reduced dimensionality may improve the performance of INB because INB does not natively scale well to high-dimensional data.

---

**Algorithm 2** Fed-1D-Barycenter

---

**Input:** Samples from the $M$ domain distributions $(\boldsymbol{z}_1, \boldsymbol{z}_2, \ldots, \boldsymbol{z}_M)$, weight vector $\boldsymbol{w}$
**Output:** Estimated invertible alignment maps $(t_1, t_2, \cdots, t_M)$
    {Clients} Estimate the 1D CDF of $Z_m$: $F_m = \text{HistogramDensityEstimation}(\boldsymbol{z}_m)$
    {Clients $\rightarrow$ Server} $F_m$
    {Server} Estimate the inverse CDF of barycenter $F_{bary}^{-1} = \sum_m w_m F_m^{-1}$
    {Server $\rightarrow$ Clients} $F_{bary}^{-1}$
    {Clients} Construct invertible alignment map $t_m = F_{bary}^{-1} \circ F_m$
    **return** $(t_1, t_2, \cdots, t_M)$

---

## 2.6 COMMUNICATION COST ANALYSIS OF FEDINB

We will now discuss the communication complexity of FedINB in terms of the number of parameters to be transmitted from each client to the server. Let $B$ be the number of histogram bins for the CDF of local slices and the inverse CDF of the barycenter in Algorithm 2, and let $V$ as the number of histogram bins we use for VW histograms in Algorithm 1. At each iteration at each layer of INB, Fed-multi-max-K-SW requires us to transmit slices of samples, empirical barycenter samples, projection matrix and gradient. Thus, the number of parameters needed to be transmitted is $\mathcal{O}(K(n + d))$

per iteration of Fed-multi-max-K-SW. At each layer of INB, the total number of parameters being transmitted for Fed-multi-max-K-SW is $\mathcal{O}(JK(n + d))$, where we assume that the number of backtracking iterations is at most a small constant more than the maximum number of iterations $J' = \mathcal{O}(J)$. For Fed-Bary (applying Algorithm 2 to $K$ dimensions simultaneously), we need to communicate histogram bins between the client and the server. The number of parameters being transmitted is $\mathcal{O}(KB)$. After applying VW histograms and the simplification for backtracking line search, the total communication cost of each layer of INB reduces to $\mathcal{O}(JK(V + d))$ per layer (assuming $B \ll JV$, which is true in practice). Thus, the communication is linear in terms of all the relevant parameters, and reducing the dimensionality $d$ with an autoencoder will immediately provide communication benefits for high-dimensional data. More details can be found in Appendix C.

## 3 RELATED WORK

Beyond the translation models introduced in Section 1.1, we discuss other related works here.

**Federated learning**  Konevcný et al. (2016) originally proposed the FL framework for promoting data privacy (Shokri & Shmatikov, 2015; Azam et al., 2022b) and enhancing communication efficiency (Wang et al., 2020). Recent years have seen an increasing interest in the adaptation of FL frameworks to settings of non-IID data with respect to the class labels. For example, Zhao et al. (2018) propose sharing a subset of client data to create a global data pool that reduces the gradient divergence across different clients with non-IID data. Another line of work (Lin et al., 2021) proposes to tackle client data heterogeneity by incorporating distributed average model consensus within a semi-decentralized architecture. However, these works do not consider the larger set of complex data heterogeneity settings that include a shift in input data distribution (or domain distribution) across clients.

**Domain generalization**  Domain generalization is an application of domain translation that targets at training models that can generalize to unseen distributions. While most existing methods require access to data from all domains in the training, there are several methods that can be used as regularization for training at each local client. For example, Carlucci et al. (2019) regularize the model's training by solving solving jigsaw puzzles on each image. Zhang et al. (2020) apply deelply stacked transformation on the source domains to simulate the possible domain shift. Another main branch of domain generalization is to learn domain-invariant features. DIRT (Nguyen et al., 2021) explicitly employs a pretrained domain translation model by minimizing the following objective:

$$\frac{1}{M} \sum_{m=1}^{M} \mathbb{E}_{p_m(x,y)}[\ell(y, g(x)) + \frac{1}{M} \sum_{m'=1}^{M} \text{dis}(g(x), g(f_{m \to m'}(x)))] \tag{4}$$

where $g(x)$ is an encoder trying to find the domain-invariant feature and $\ell$ is the classification loss given the representation $g(x)$ in the encoded space. In their paper, they use StarGAN as $f$, and as discussed in Section 1, we propose to use FedINB for federated domain generalization.

**Federated domain generalization**  Understandably, there are few works that consider the problem of domain generalization [1] in a federated setting owing to its high complexity as discussed in Section 1. Liu et al. (2021) solve the lack of domain knowledge across clients by creating a common bank of amplitude spectrum of data which is aggregated by accumulating the amplitude spectrums obtained by application of Fast-Fourier Transforms (FFTs) on the client data. While this trick leads to significant performance gains, it significantly reduces privacy because the clients are essentially sharing half of their datasets (i.e., their amplitude spectrums) with all other clients. A more detailed discussion on privacy can be found in Appendix D. Besides, the proposed method by Liu et al. (2021) is specifically developed for image segmentation tasks and it is not clear how can the same be extended to a more general task and other data modalities such as text, tabular or other modalities.

---

[1]Note that this is different from domain adaptation. Suppose we have source domains and target domains. In DG, each source (client) has access to both data instance $x$ and class label $y$, and the goal is to generalize to target domains (new clients) where neither are observed in the training. In domain adaptation, except for $x$ and $y$ in the source domain, we also observe the data instance $x$ of the target domain in the training.

## 4 EXPERIMENTS

Before presenting our experimental results, we first introduce a naming convention to describe many different setups of FedINB. For simplicity, we will remove the "Fed" prefixes and call it INB since all experiments are done in a federated setting. With the autoencoder, the model is called AEINB if using a shared autoencoder for all domains, and IndAEINB if using autoencoders trained on each domain separately. When using VW histograms to replace the transmission of real data in Fed-multi-max-K-SW, we call them HistINB, HistAEINB, HistIndAEINB separately. The number of layers $L$, number of dimensions $K$ and max number of iterations (in Fed-multi-max-K-SW) $J$ are three key hypeparameters to tune when building an INB model. We will represent them in the format $L$-$K$-$J$. As an example, "HistIndAEINB-L10-K10-J100" refers to an INB model with $L = 10, K = 10, J = 100$, and it incorporates an autoencoder trained locally and uses VW histograms when optimizing for $\boldsymbol{\theta}$.

**Datasets** Following the setup in Zhou et al. (2022), we test FedINB on Rotated MNIST and FashionMNIST (Ghifary et al., 2015). For the federated domain translation experiment, there are 5 clients participating in the training and each has data from one of domains $0, 15, 30, 45, 60$, where the number represents the degree of counter-clockwise rotation and 0 means the original images. For the federated domain generalization task, we use the translation model trained on domains $0, 15, 30, 45, 60$. We test the classification model's generalizing ability to domains $75$ and $90$.

**Metrics** For the federated domain translation experiment, we use the empirical Wasserstein Distance (WD) and FID score (Heusel et al., 2017) between the original samples and translated samples as evaluation metric. Wasserstein Distance is computed as $\text{WD} = \frac{1}{M^2} \sum_m \sum_{m'} \widehat{WD}(\boldsymbol{x}_m, f_{m' \to m}(\boldsymbol{x}_m))$ where each $\widehat{WD}$ is computed with the Sinkhorn algorithm (Cuturi, 2013). The FID score is computed as $\text{FID} = \frac{1}{M^2} \sum_m \sum_{m'} FID(\boldsymbol{x}_m, f_{m' \to m}(\boldsymbol{x}_m))$. The WD and FID metrics reported are the average of 10 classes of digits since we are conducting class-wise translation. All FID score results can be found in Appendix F. To compare communication cost, we consider the number of parameters needed to be transmitted between one client and the server. We also measure the performance of our federated translation models by using it for downstream federated DG tasks.

### 4.1 PRACTICAL IMPROVEMENT OF FEDINB

Due to space limitations, we only show one figure for each investigation with Rotated MNIST. More figures with different setups of INB can be found in Appendices F.1 and F.2. Figures with Rotated FashionMNIST, qualitative results and investigation of the FedINB optimization can be found in Appendices F.3, F.4 and F.6 respectively.

**Autoencoder.** We investigate how adding autoencoders as part of the translation model can help with translation performance and communication cost. As can be seen in Figure 2a, both AEINB and IndAEINB achieve lower WD and lower communication cost than INB while other parameters are kept the same. Additionally, we show that by using an independent autoencoder for each domain separately, we can obtain lower WD for the same communication cost while simultaneously reducing the work of training a federated shared autoencoder.

**Insensitivity to $J$.** In the original INB paper, the authors set $J = 200$ for MNIST (LeCun & Cortes, 2010) and FahsionMNSIT (Xiao et al., 2017). This could lead to a high communication cost due to frequent transmission of data (even if the data size is small). We empirically show that even when the optimization algorithm does not converge and we stop early, it will not affect the final performance much. One explanation for this is that the projection matrix found suffers less from overfitting. As shown in Figure 2b, after 10 layers of INB, the model with $J = 50, 100, 200$ converge to a similar point. Even the model with $J = 30$ almost begins to converge at the same point; only when $J$ is decreased to a quite small value like 10 will the convergence of INB require more layers.

**VW histograms.** As shown in Figures 2c, 2d and 2e, when using VW histograms and decreasing $V$, the communication cost is significantly reduced. In comparison to the number of samples being transmitted by INB and IndAEINB ($n = 10000$), $V = 500$ can tolerate the transmission of substantially less samples while achieving similar performance.

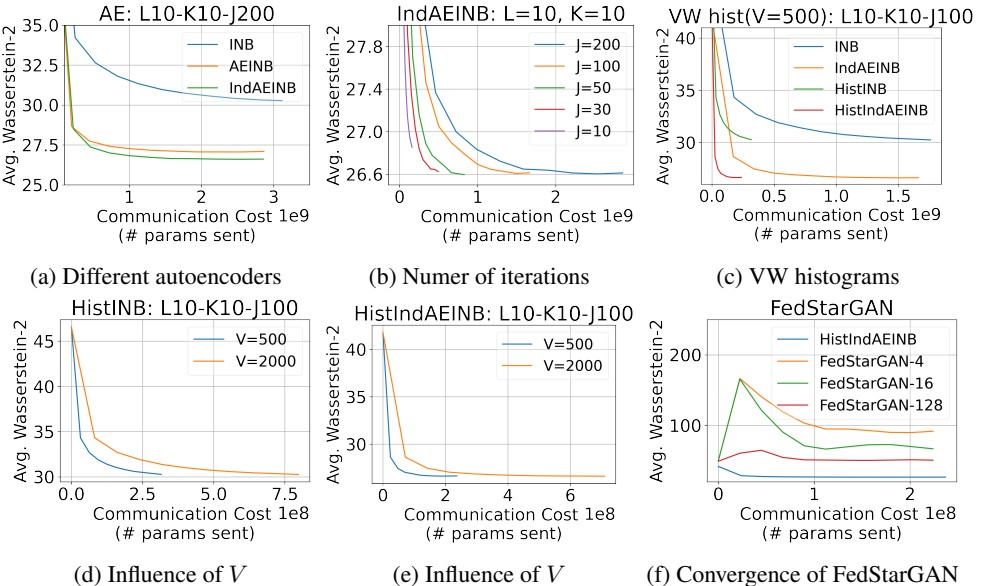

Figure 2: Wasserstein Distance (lower is better) as communication cost increases for Rotated MNIST. (a) Different autoencoders used for INB. (b) Different number of inner iterations used for IndAEINB. (c) Using Variable-Bin-Width (VW) histograms for INB. (d) HistINB: different number of histogram bins used for VW histograms. (e) HistIndAEINB: different number of histogram bins used for VW histograms. (f) Failure of FedStarGAN. The INB used here is HistIndAEINB-L10-K10-J100 with $V = 500$. The starting points of HistAEINB and FedStarGAN differ because we begin tracking the WD after applying autoencoders to INB (details can be found in Appendix E).

## 4.2 COMMUNICATION-EFFICIENT DOMAIN TRANSLATION

We compare with FedStarGAN[2] to demonstrate that our translation model is much more efficient in terms of both communication and computational cost. We empirically show that the training of StarGAN fails in the federated setting. We adapt StarGAN to the federated setting by training individual StarGAN models on each client (using the local data, i.e., single domain) and synchronizing the client generator and discriminator after a certain number of mini-batches (controlled using the `sync_step` hyperparameter) using FedAvg. We try various `sync_step` sizes to fairly evaluate the federated extension of StarGAN (FedStarGAN). As a note, FedStarGAN-128 means using FedAvg to train StarGAN and synchronizing after each 128 mini-batches (`sync_step`=128).

As shown in Figure 3, at a fixed communication cost, even though FedStarGAN can generate images with good quality, it fails to translate it to the target domain, i.e., it can only reconstruct the input but not translate to a rotated version of the image. This is in line with our observation during training that while FedStarGAN can optimize for reconstruction loss, it does not converge over the domain adversarial loss. Moreover, in Figure 2f, we observe that FedStarGAN-128 (which converges fastest) converges to its starting point, justifying our argument that it can only reconstruct images. Additionally, it is important to note that INB takes around 5 minutes to finish training on a single RTX A5000 GPU while FedStarGAN takes around 2.5 hours to generate good samples on a single Tesla P100 GPU (notice it can only reconstruct samples rather than translate samples).

## 4.3 FEDERATED DOMAIN GENERALIZATION VIA PSUEDO TRANSLATED DOMAIN DATA

Once the translation models have been trained in a federated way, clients can exchange their models so that each can translate to every other domain. This enables the training of DIRT (Nguyen et al., 2021) for DG using these translation models (that were also learned via FL). Specifically, we combine FedAvg and DIRT – named FedDIRT – where each client $m$ minimizes the following local objective:

$$\mathbb{E}_{p_m(x,y)}[\ell(y, g(x)) + \tfrac{1}{M} \sum_{m'=1}^{M} \text{dis}(g(x), g(f_{m \to m'}(x)))], \qquad (5)$$

---

[2]StarGAN is trained with the standard FedAvg algorithm (McMahan et al., 2017). To improve performance, we synchronize quite frequently – after certain mini-batches rather than certain epochs.

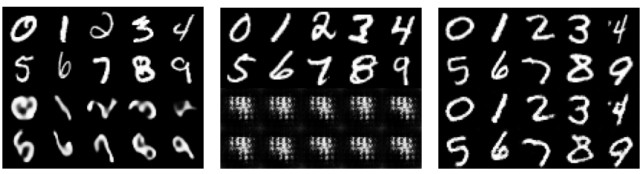

(a) INB      (b) FedStarGAN-16    (c) FedStarGAN-128

Figure 3: Qualitative results for federated domain translation using INB and FedStarGAN when around $2 \times 10^8$ parameters are transmitted. The first two rows are original samples from domain 0 and the last two rows are samples translated to domain 60. (a) *Our HistIndAEINB-L10-K10-J100 algorithm*: It translates the digits to the expected domain. (b) *FedStarGAN after 160 iterations and Fedavg every 16 steps*: It fails to generate legible digits. (c) *FedStarGAN after 1280 iterations and FedAvg every 128 steps*: The generated samples are well reconstructed but are not translated.

Table 1: Classification accuracy on unseen domains for Rotated MNIST. For each target domain, we consider aggregating the model after each 1 mini-batch and 100 mini-batches. The mean and standard deviation are taken over 4 runs. For FedDIRT, we use HistIndAEINB with $J = 100$ and $V = 500$.

| Model | 75 (1-batch) | 75 (10-batch) | 75 (100-batch) | 90 (1-batch) | 90 (10-batch) | 90 (100-batch) |
|---|---|---|---|---|---|---|
| FedDIRT(L20-K10) | **92.2 ± 0.3** | 89.3 ± 4.2 | **91.9 ± 0.9** | **69.8 ± 0.7** | 70.4 ± 2.2 | **69.9 ± 0.3** |
| FedDIRT(L20-K20) | **92.2 ± 1.3** | **90.8 ± 1.2** | 91.4 ± 0.6 | 69.4 ± 0.8 | **71.1 ± 1.3** | 69.8 ± 1.5 |
| FedAvg | 85.2 ± 0.7 | 85.1 ± 0.5 | 80.1 ± 2.3 | 63.8 ± 2.1 | 63.6 ± 0.8 | 55.6 ± 2.0 |

Table 2: Classification accuracy on unseen domains for Rotated FashionMNIST.

| Model | 75 (1-batch) | 75 (10-batch) | 75 (100-batch) | 90 (1-batch) | 90 (10-batch) | 90 (100-batch) |
|---|---|---|---|---|---|---|
| FedDIRT(L20-K10) | **65.8 ± 1.3** | 63.6 ± 0.9 | 63.0 ± 1.0 | 18.2 ± 0.9 | 18.1 ± 0.4 | 18.8 ± 0.6 |
| FedDIRT(L20-K20) | 65.5 ± 1.8 | **64.5 ± 1.5** | **63.5 ± 1.3** | **19.0 ± 0.3** | **18.5 ± 1.0** | **18.9 ± 0.4** |
| FedAvg | 51.9 ± 2.9 | 50.4 ± 1.2 | 40.4 ± 4.1 | 13.7 ± 1.6 | 14.6 ± 1.8 | 13.1 ± 0.9 |

and the server aggregates the model after certain mini-batches. Note that this local objective only uses data from the $m$-th client but leverages the shared FedINB translation models for the DIRT regularization. We run FedAvg without any domain translation regularization loss term as our baseline. For fairness, we use the same CNN structure for both models. Table 1 and Table 2 give the results for training the domain translation model on domains 0,15,30,45,60 and using it to train a model that can generalize to unseen domains 75 and 90. More results can be found in Appendix F. We can observe that regularization from domain translation model significantly improves the model's ability to generalize to unseen domains. Additionally, we notice that as we increase the synchronization period (i.e., decrease the FL aggregation frequency), which is an important objective in communication-constrained FL systems (Lin et al., 2021), FedDIRT achieves a similar performance as before while the performance of FedAvg drops significantly. The regularization helps make the federated training more stable. This can also be justified by the higher standard deviation of most FedAvg runs.

## 5   Discussion and Conclusion

Despite the theoretical and empirical advantages of FedINB, some limitations should be noted for future work. For example, it is difficult for FedINB to translate high dimensional images: a possible solution is to further reduce the dimension via a (pretrained) deep neural network in the federated domain generalization task. Also, we have not provided a theoretical privacy guarantee for FedINB, which could possibly be accomplished by utilizing DP-SGD (Abadi et al., 2016) for Fed-multi-max-K-SW. More discussion on these points can be found in Appendix C.5.

In this paper, we proposed a federated domain translation approach (FedINB) that can mitigate non-IID issues in federated learning tasks. Our model can handle the harder task of conditional shift in comparison to most existing FL methods. We then proposed several improvements to FedINB that decrease communication costs for practical resource-constrained FL systems. We empirically showed that our translation model performs substantially better than FL versions of standard translation models (StarGAN). Finally, as an application, we demonstrated that combining FedINB with SOTA domain generalization methods leads to strong performance in federated domain generalization.

ACKNOWLEDGMENTS

Z.Z. and D.I. acknowledge support from NSF (IIS-2212097) and ARL (W911NF-2020-221). S.A. and C.B. acknowledge support from NSF (CNS-2146171) and ONR (N000142212305).

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

## A    OVERVIEW

We have organized our appendix as follows:

- Appendix B includes the proof of Remark 1.
- Appendix C includes more details about the INB algorithm, its implementation, limitations and future works.
- Appendix D includes more discussion on privacy.
- Appendix E includes experiment and implementation details.
- Appendix F includes more experiment results.

## B    PROOF OF REMARK 1

Before starting the proof of Remark 1, let us first introduce the following definition and lemma to deal with the sorting operation in the objective.

**Definition 2.** *The sorting permutation matrix $P_Z \in \mathbb{R}^{n \times n}$ associated with a vector $Z \in \mathbb{R}^{1 \times n}$ is defined as $ZP_Z = \text{sorted}(Z)$ such that $(ZP_Z)_i \leq (ZP_Z)_{i+1}, i = 0, \dots, n-1$.*

**Lemma 1.** *Let $A \in \mathbb{R}^{1 \times d}, X \in \mathbb{R}^{d \times n}$, and denote the sorting permutation matrix of $AX$ as $P_{AX}$. We have:*

$$\frac{\partial(AXP_{AX})}{\partial A} = (XP_{AX})^T$$

*Proof.* Note that $AXP_{AX} = \text{sort}(AX)$ where sort is the sorting function. With this, we first note that the Jacobian of sort with respect it's inputs is equal to the permutation matrix that sorts that input.

$$J_{\text{sort}}(z) = \frac{\partial \text{sort}(z)}{\partial z} = P_z^T$$

where $v = P_z z$ is sorted, i.e., $v_i \leq v_{i+1}, \forall i < n$ (Blondel et al., 2020).[3] Now we want the following:

$$\frac{\partial \text{sort}(AX)}{\partial A} = \frac{\partial \text{sort}(AX)}{\partial(AX)}\frac{\partial AX}{\partial A} = P^T X^T = (XP)^T.$$

□

Now we prove Remark 1.

*Proof of Remark 1.* This remark in essence states that when we compute the gradient of sum of the loss among all clients with respect to the projection matrix, the empirical barycenter could be regarded as a constant (zero gradient). Intuitively, the proof relies on the fact that we are using squared distance, and the empirical barycenter can be seen as a mean value over domains.

Let $A \in \mathbb{R}^{1 \times d}, X_i \in \mathbb{R}^{d \times n}, i = 1, \dots M$ and define the loss term $L$ as

$$L \triangleq \sum_{i=1}^M \left\| AX_i P_{AX_i} - \frac{1}{M}\sum_{j=1}^M (AX_j P_{AX_j}) \right\|^2.$$

Note that $L$ is equivalent to $\sum_{k=1}^n \sum_{i=1}^M |(AX_i)_{[k]} - \frac{1}{M}\sum_j (AX_j)_{[k]}|^2$. Then define the loss $L^C$ as

$$L^C \triangleq \sum_{i=1}^M \|AX_i P_{AX_i} - C\|^2$$

---

[3]Blondel, Mathieu, et al. "Fast differentiable sorting and ranking." International Conference on Machine Learning. PMLR, 2020.

where $C \triangleq \frac{1}{M} \sum_{j=1}^{M} AX_j P_{AX_j}$ but will be regarded *as a constant* when computing the gradient of $L^C$. And for simplicity, we denote $X_i' \triangleq X_i P_{AX_i}$. We have:

$$
\begin{aligned}
\frac{\partial L}{\partial A} &= 2 \sum_i (AX_i P_{AX_i} - \frac{1}{M} \sum_{j=1}^{M} AX_j P_{AX_j}) \frac{\partial}{\partial A} (AX_i P_{AX_i} - \frac{1}{M} \sum_j AX_j P_{AX_j})^T \\
&\overset{(1)}{=} 2 \sum_i (AX_i' - \frac{1}{M} \sum_{j=1}^{M} AX_j')(X_i' - \frac{1}{M} \sum_{j=1}^{M} X_j')^T \\
&= 2 \sum_i AX_i' X_i'^T - \frac{4}{M} \sum_i \sum_j AX_i' X_j'^T + \frac{2}{M} \sum_i \sum_j AX_i' X_j'^T \\
&= 2 \sum_i AX_i' X_i'^T - \frac{2}{M} \sum_i \sum_j AX_i' X_j'^T
\end{aligned}
$$

where (1) uses Lemma 1. Additionally,

$$
\begin{aligned}
\frac{\partial L^C}{\partial A} &= 2 \sum_i (AX_i P_{AX_i} - C) \frac{\partial}{\partial A} (AX_i P_{AX_i}) \\
&= 2 \sum_i (AX_i P_{AX_i} - C) X_i'^T \\
&= 2 \sum_i (AX_i' - \frac{1}{M} \sum_{j=1}^{M} AX_j') X_i'^T \\
&= 2 \sum_i AX_i' X_i'^T - \frac{2}{M} \sum_i \sum_j AX_i' X_j'^T \\
&= \frac{\partial L}{\partial A}
\end{aligned}
$$

$\square$

## C MORE DISCUSSION OF THE FEDINB ALGORITHM

### C.1 ORIGINAL INB AND FEDINB

Zhou et al. (2022) propose the Iterative Naïve Barycenter (INB) algorithm to align high dimensional distributions via decomposing the problem into simpler 1D alignment problems with closed-form solutions. Specifically, they first reduce the dimension via a orthogonal projection matrix found by minimizing maximum K-sliced Wasserstein divergence as shown in Equation (2). Informally, this objective finds orthogonal directions where the empirical Wasserstein-2 distance to the empirical Wasserstein barycenter is maximized—where both the empirical Wasserstein-2 distance and barycenter can be solved in closed-form only using 1D sorting. They adopt the optimization approach of (Dai & Seljak, 2021) to optimize over the Stiefel manifold of orthonormal matrices via projected gradient descent with backtracking line search. Then, along each dimension, they solve the 1D Wasserstein Barycenter and find the corresponding mapping to the Barycenter. By iteratively finding the projection matrix and 1D mapping to the Barycenter, they construct a deep translation model between distributions. More details about INB can be found in (Zhou et al., 2022).

In our paper, we propose adapting INB for federated domain translation. The main FedINB algorithm employing Algorithms 1&2 is given in Algorithm 3.

### C.2 VW HISTOGRAMS

One key improvement made for FedINB is to use VW histograms to approximate the empirical slice Wasserstein distance objective in Algorithm 1. It is noteworthy that if the number of histogram bins is the same as number of samples, the FedINB with VW histograms becomes equivalent to INB. In this case, the quantiles sent would just be the samples and each of them has uniform density. As we

---

**Algorithm 3** Federated Iterative Naïve Barycenter

---

**Input:** Samples from $M$ clients $\boldsymbol{x}_1, \boldsymbol{x}_2, \ldots, \boldsymbol{x}_M$, number of directions $K$, number of iterations/layers $L$

**Output:** Estimated domain translation models $(T_1, T_2, \ldots, T_M)$

  {Client} $T_m^{(0)} \leftarrow \mathrm{id}, \quad \forall m = \{1, \ldots, M\}$
  **for** $\ell = \{1, 2, \ldots, L\}$ **do**
    {Client} $\forall m, \boldsymbol{z}_m \leftarrow T_m(\boldsymbol{x}_m)$
    {Server, Client} $\boldsymbol{\theta} \leftarrow$ Fed-multi-max-K-SW $((\boldsymbol{z}_1, \ldots, \boldsymbol{z}_M), K)$ {Algorithm 1}
    **for** $k = \{1, \ldots, K\}$ **do**
      {Client} $\forall m, \boldsymbol{z}'_m = \boldsymbol{\theta}_k^T \boldsymbol{z}_m$
      {Server, Client} $t_{1,k}, \ldots, t_{M,k} = $ 1D-Barycenter$(\boldsymbol{z}'_1, \ldots, \boldsymbol{z}'_M)$ {Algorithm 2}
    **end for**
    {Client} $\forall m, t_m \leftarrow [t_{m,1}, \ldots, t_{m,K}]$
    {Client} $\forall m, T_m(\boldsymbol{x}) \leftarrow \boldsymbol{\theta} t_m(\boldsymbol{\theta}^T T_m(\boldsymbol{x}))$
  **end for**
  **return** $(T_1, T_2, \cdots, T_M)$

---

show empirically in Section 4 and Appendix F, even if the number of bins is smaller than the number of samples, this approximation will not hurt the final performance and can reduce the communication cost.

### C.3 COMMUNICATION SIMPLIFICATION FOR BACKTRACKING LINE SEARCH

When optimizing the projection matrix $\boldsymbol{\theta}$, the original multi-max-K-SW algorithm does backtracking line search which requires recomputing the loss with an updated $\boldsymbol{\theta}$ where only the learning rate is changed. Thus, given a fixed maximum number of iterations $J$, we may need to run extra iterations $J'$. At each backtracking iteration, we need to compute slices and the empirical barycenter with updated $\boldsymbol{\theta}$ to recompute the loss so that we can determine whether to accept this update. We propose to locally record the change of $\boldsymbol{\theta}$ so that if backtracking line search determines a reduction in the learning rate, we can reverse-engineer the new updated $\boldsymbol{\theta}$ without explicit synchronization of the new $\boldsymbol{\theta}$.

### C.4 COMPUTATIONAL COMPLEXITY AND COMMUNICATION COST

*Computational complexity.* The complexity of the maximization (Algorithm 1) is $\mathcal{O}((J + J')(nMK(d + \log n) + K^2 d + K^3))$, where $J, J', M, n, d, K$ are the number of iterations for Fed-multi-max-K-SW , extra iterations for backtracking line search, domains, samples per domain, dimensions, and latent dimensions, respectively. If we use VW histograms, the complexity reduces to $\mathcal{O}((J + J')(VMK(d + \log V) + K^2 d + K^3))$. The complexity of the minimization (applying Algorithm 2 for each latent dimension independently) is $\mathcal{O}(nMK)$.

*Communication cost.* We gave an approximated communication cost in the main paper. Here, we provide a more exact estimation. At each iteration of INB, for each client, the communication cost of the maximization (Algorithm 1) is $\mathcal{O}((J + J')K(n + d))$ and $\mathcal{O}((J + J')K(V + d))$ for FedINB and FedINB with VW histograms respectively. The communication cost of minimization (applying Algorithm 2 to $K$ dimensions simultaneously) is $\mathcal{O}(K(B_1 + B_2))$ where $B_1$ and $B_2$ are the number of histogram bins for the CDF of local slices and inverse CDF of the Barycenter.

### C.5 LIMITATION AND FUTURE WORKS

Similar to the original INB, a limitation of FedINB is that it does not perform well for high dimensional image datasets. One of the reasons could be that FedINB is a greedy algortihm. On one hand, this leads to benefits in computation and communication. On the other hand, this does not guarantee we find the global optimum of the alignment problem. From a theoretical perspective, as pointed out in Zhou et al. (2022), the best we can guarantee is that at each iteration, given the current $\boldsymbol{\theta}$, the local alignment map $t_m$ is optimal. Thus, in optimizing $\boldsymbol{\theta}$, especially considering the addition of VW histogram and reducing the maximum number of iterations $J$ due to communication concerns, the

empirical maximization of multi-max-K-SW may not find the true maximum. But, as we empirically show in the experiment section, neither leads to drop in performance unless $J$ or the maximum number of histogram bins $V$ is really small. To further improve this fundamentally, we could investigate more on the optimization on the manifold of orthonormal matrices.

To overcome this, we provide practical improvements such as using autoencoders to improve the performance. Also, FedINB could employ some specific tricks for images such as patch-based hierarchical structure in SINF (Dai & Seljak, 2021) (though we do not leverage these image-specific tricks in this paper). Instead, we focus on a more general domain-agnostic method that does not assume any image-based structure in the model design. Since we focus on the more general case, Wasserstein Distance is a more reasonable metric as it is domain-agnostic and not specific to images. Furthermore, for the application of federated DG on high dimensional image datasets, we could share a pretrained deep neural networks, which is known to be good at extracting meaningful features. Then we train the federated translation model in the intermediate representation. In this case, we don't need this preliminiary transformation to be invertible since we don't need to get the exact translated images.

Another limitation of FedINB is that we need to transmit the translation model if we want to use it for other downstream tasks such as federated domain generalization. While this is more desirable than directly transmitting data in comparison to models such as ELCFS (Liu et al., 2021), we propose a possible method to mitigate this issue altogether. For federated domain generalization, we can utilize the shared space learned by FedINB for applying FedDIRT. Specifically, each client $m$ will minimize the following local objective:

$$\mathbb{E}_{p_m(x,y)}[\ell(y, g(x)) + \text{dis}(g(x), g(T_m(x)))] . \tag{6}$$

The key difference is that the local objective on the $m$-th client would only need its own translation model to the shared space $T_m$ rather than needing $T_{m \to m'}$, which would require the translation models from other clients. The intuition is that even though the shared space learned by FedINB is not the actual Wasserstein Barycenter, it may still keep the geometric structure of each domain and it may share the same invariant representation as the distribution of each domain. Hence, the second term may still serve as a good regularizer for DIRT. This has the potential to further improve privacy preservation (because the client-specific translation models would not be shared) and communication cost (because the translation models would not have to be transmitted). However, we leave a formal investigation of FedDIRT with the shared space of FedINB for future work.

Regarding privacy, we discussed about why sorting and VW histograms could improve privacy. More detailed discussion and limitation are in the next section.

## D    DISCUSSION ON PRIVACY

In this section, we will discuss how our FedINB with VW histograms is better than ELCFS (Liu et al., 2021) and original FedINB in terms of privacy. However, we want to clarify upfront that we do not claim our model satisfies any strict privacy metric from the machine learning and federated learning areas such as differential privacy (DP) (Dwork, 2006). [4] Rather, we only seek to demonstrate better privacy *in comparison* to prior and naïve approaches. Regarding DP, we provide a discussion at the end of this section and will leave more formal analysis and careful investigation for future works.

In what follows, we will compare methods from the perspective of membership inference attacks. Positive membership privacy is concerned about an adversary's ability to infer that an entity participated the training as defined in (Li et al., 2013, Definition 3.1)[5]. As a running example for clarity, we will assume the hospital scenario where each client is a hospital and each observation is a patient's data (e.g., cell images), but the following arguments could be made in other privacy scenarios as well. In our case, we make two assumptions: 1) The adversary has the original data of one patient (which may or may not be one of the training examples at one of the hospitals) and 2) the adversary can view the information sent to and from the central server. The membership inference question is: *Given these assumptions, can the adversary determine whether the patient's data was included in the training data or not?*

---

[4]Cynthia, Dwork. "Differential privacy." Automata, languages and programming (2006): 1-12.

[5]Li, Ninghui, et al. "Membership privacy: a unifying framework for privacy definitions." *ACM SIGSAC Conference on Computer & Communications Security.* 2013.

We first discuss membership attacks for ELCFS and then discuss them with respect to the privacy improvements of FedINB. ELCFS first does a fast Fourier transform of all data in each client to get the amplitude spectrum and phase spectrum. Then they transmit the low-frequency component of the amplitude spectrum to the server to form a distribution bank. In the training, each client acquires those low-frequency spectrums from other clients to create pseudodata from different domains. We believe this operation significantly violates the positive membership privacy. An adversary can compute the fast Fourier transform of the original data (assumption 1) and check whether its low-frequency component of amplitude spectrum is in the distribution bank, which is shared to each client in the training (assumption 2).[6]

The sorting of slices of data *before* sending to the server in Algorithm 1 makes membership attacks more difficult than if the sorting was done on the server. Intuitively this is because sorting destroys joint sample information. First, consider the case where sorting is done on the server. We argue that an membership inference attack would be almost trivial in this case. The adversary could project the original data (assumption 1) using the current projection matrix (assumption 2) and match this projected data with *a whole row* in the projected training examples that are transmitted during training (assumption 2). If there is not a whole-row match, then the patient is not in the training data. If there is a whole-row match, then the patient is more likely to be in the training data, where $K$ determines the amount of precision—i.e., if $K = d$ and all data instances are unique, then this proves that the patient is in the training data, but if $K = 1$, there may be multiple patients that have the same projected value. On the other hand, if sorting is applied at the clients *before* sending, then the adversary must attempt to determine a match by matching on each feature independently because the sorting indices are hidden. Thus, the adversary cannot match on whole rows anymore. The amount of actual privacy will depend on the non-uniqueness of individual values.[7] Furthermore, sending quantiles instead of real samples improves the privacy with respect to membership attacks because directly sending projected samples may hurt privacy. With VW histograms, most samples' exact information would be impossible to reconstruct (as histograms are lossy compressions of data samples) and trivial membership inference could be prevented. In some cases, such as the min and max values, membership attacks could still be possible but they are significantly more difficult for VW histograms than merely sending samples. Again, we do not claim DP guarantees but discuss below how these could be added on top of our framework.

Finally, we provide several possibilities to ensure DP on top of the proposed FedINB. First, our method could be extended to leverage DP-SGD (Abadi et al., 2016) [8] for the optimization of projection matrix by merely clipping the gradient values and adding noise before locally updating the parameters. Regarding transmission of VW histograms for the barycenter calculations, our method could also be extended to compute differentially private histograms by adding noise to the histogram estimates as is suggested in Example 3.2 of Dwork & Roth (2014) [9]. Finally, we could use standard DP techniques for composing multiple DP mechanisms (in this case DP-SGD and and DP histograms) into a joint DP mechanism that has DP guarantees (Dwork & Roth, 2014). We leave a more careful investigation of DP methods in this context for future work.

## E    EXPERIMENTAL DETAILS

### E.1    FEDERATED DOMAIN TRANSLATION

**Metrics** As discussed in the main paper, we choose WD and FID as our two metrics, which gives a fair comparison against baselines. They are computed as WD $= \frac{1}{M^2} \sum_m \sum_{m'} \hat{WD}(\boldsymbol{x}_m, f_{m' \to m}(\boldsymbol{x}_{m'}))$ and FID $= \frac{1}{M^2} \sum_m \sum_{m'} FID(\boldsymbol{x}_m, f_{m' \to m}(\boldsymbol{x}_{m'}))$ (fixed

---

[6]For common Deep Learning datasets such as high-dimensional images, the low-frequency component of amplitude spectrum is very likely to be unique given the high signal-to-noise ratio of real images.

[7]If the feature values are discrete, then there is likely to be many overlapped values and membership attacks will be difficult. If the feature values are continuous, then each value will be theoretically unique but adding some small Laplace noise, as is standard in differential privacy, would ensure that values are no longer unique.

[8]Abadi, Martin, et al. "Deep learning with differential privacy." *ACM SIGSAC Conference on Computer and Communications Security.* 2016.

[9]Dwork, Cynthia, and Aaron Roth. "The algorithmic foundations of differential privacy." Found. Trends Theor. Comput. Sci. 9.3-4 (2014): 211-407.

a typo in the main paper). WD is computed with the Sinkhorn algorithm with maximum iterations set to 100 and $\epsilon = 0.0001$. At the starting point, $f$ is the identity function.

**Dataset** For training, we use 10,000 samples from the MNIST and FashionMNIST training set as the dataset of domain 0, where each class has 1000 samples. Then we use all samples to generate 10,000 samples for all other training domains (15,30,45,60). So, the total size of training data is 50,000. For evaluation of FedINB, we also use 10,000 samples from the MNIST and FashionMNIST test set, and create other samples in the same way. So the total size of test data is also 50,000. For evaluation of FedStarGAN, due to high computational cost, we only use 2,000 samples to create test dataset so the total size of test data is 10,000. We find that the difference of evaluation score caused by different sizes of test dataset is negligible especially, for the WD score, and will not affect our conclusion in the paper.

**FedINB with autoencoders** In both AEINB and IndAEINB, we use the same autoencoder structure. For MNIST, the encoder is composed of [`nn.Conv2d(1, 16, 3, padding=1)`, `nn.ReLU(inplace=True)`, `nn.MaxPool2d(2)`, `nn.Conv2d(16, 8, 3, padding=1)`, `nn.ReLU(inplace=True)`, `nn.MaxPool2d(2)`] where `nn` represents `torch.nn` in PyTorch. The decoder is composed of [`nn.Conv2d(8, 8, 3, padding=1)`, `nn.ReLU(inplace=True)`, `nn.UpsamplingBilinear2d(scale_factor=2)`, `nn.Conv2d(8, 16, 3, padding=1)`, `nn.ReLU(inplace=True)`, `nn.UpsamplingBilinear2d(scale_factor=2)`, `nn.Conv2d(16, 1, 3, padding=1)`, `nn.Sigmoid()`] where the last Sigmoid activation is used to map the output to the range of $[0, 1]$. For FashionMNIST, the encoder is composed of [`nn.Conv2d(1, 8, 3, stride=2, padding=1)`, `nn.ReLU(True)`, `nn.Conv2d(8, 16, 3, stride=2, padding=1)`, `nn.BatchNorm2d(16)`, `nn.ReLU(True)`, `nn.Conv2d(16, 32, 3, stride=2, padding=0)`, `nn.ReLU(True)`]. The decoder is composed of [`nn.ConvTranspose2d(32, 16, 3, stride=2, output_padding=0)`, `nn.BatchNorm2d(16)`, `nn.ReLU(True)`, `nn.ConvTranspose2d(16, 8, 3, stride=2, padding=1, output_padding=1)`, `nn.BatchNorm2d(8)`, `nn.ReLU(True)`, `nn.ConvTranspose2d(8, 1, 3, stride=2, padding=1, output_padding=1)`, `nn.Sigmoid()`]. As pointed out in the main paper, AEINB uses a shared autoencoder that is trained on data from all training domains. In practice, we should use some federated algorithm to train a shared autoencoder while each client has access to data from one domain. However, as a test, we show that even in this case, IndAEINB achieves competitive or even better result and in the federated case, the performance of AEINB cannot be better. Hence, we only report results of AEINB using this autoencoder.

For IndAEINB, each client has its own autoencoder that is trained on the same training dataset used to train INB.

We want to clarify that when reporting scores of AEINB and IndAEINB, the first number (at number of parameters sent equal to 0) is after applying autoencoders. This is why the starting point of them is different from that of other models such as INB.

**Implementation of VW histograms** In Algorithm 1, when computing the objective, we use VW histograms to approximate the empirical Wasserstein Barycenter. At each client, we find the quantiles of sliced data (the number of quantiles is determined by $V$) and send the quantiles to the server and average them. Then we have the VW histogram density of the empirical Wasserstein Barycenter using the averaged quantiles. Finally we create a quantile of $[0, 1]$ and use the inverseCDF of the histogram density estimator to create pseudodata of the empirical Wasserstein Barycenter.

**Federated implementation of INB** To emulate a federated version of INB, we need detach the gradient of the empirical Wasserstein Barycenter in Algorithm 1. Other than that, there is not any practical difference whether we physically separate the data across different clients and simulate the transmission of data. In terms of communication cost, we compute the number of parameters being transmitted and report them.

**FedStarGAN** In order to implement FedStarGAN we start with the centralized implementation of StarGAN on RMNIST made available by the authors of DIRT. We next extend this to federated

setting by defining 5 disjoint clients (1 for each domain of RMNIST) such that each client has its own copy of generator and discriminator. The 5 source domains are distributed among the 5 clients such that each client has data from only 1 of the 5 domains. The generator and discriminator undergo periodic FedAvg. We control the aggregation frequency by define a tunable parameter `step_size` that determines the number of local update steps that client models must undergo before FedAvg aggregation and synchronization. For aggregation, we use the uniformly weighted FedAvg. Our experiments show that even 10 rounds of aggregation steps (using different values of `sync_step`) consumes more communication bandwidth than the HistIndAEINB model but underperforms in terms of WD score (Fig. 2(d)).

### E.2 Federated Domain Generalization

**Dataset** For training, we use 1000 samples from MNIST and FashionMNIST training set as the dataset of domain 0 where each class has 1000 samples. Then we use all samples to generate 1000 samples for all other training domains (15,30,45,60). So, the total size of training data is 5000. We decrease the size of the dataset to make this a harder domain generalization problem, and we retrain INB using less data. For evaluation, we use 1000 samples from MNIST and FashionMNIST training set to generate test data in new domains 75 and 90.

**Metrics** We use the accuracy of the trained classifier when employed to a new domain as the metric. Note that the INB and autoencoders are not needed after the training of classifier. As finding the most appropriate validation domain is another hyperparameter to tune, we choose to use the simplest way to report accuracy - we report the test accuracy at a fixed point. For fairness, we report the accuracy of FedDIRT and FedAvg at the same point and check that both have already converged before recording (this could be validated by Figure 14). For Rotated MNIST and all test domains, we run FedDIRT/FedAvg (1-batch) for 2000 iterations, FedDIRT/FedAvg (10-batch) for 2500 iterations and FedDIRT/FedAvg (100-batch) for 3000 iterations. For Rotated FashionMNIST and all synchronization steps, we run FedDIRT/FedAvg (Domain 75) for 1500 iterations and FedDIRT/FedAvg (Domain 90) for 3000 iterations.

**FedDIRT** For the network structure (encoder and classifier) and training hyperparameters, we modify based on the default setup in the repository of DIRT (Nguyen et al., 2021), which can be found at their public repository https://github.com/atuannguyen/DIRT. The only difference is that we change Batch Normalization to Instance Normalization. For FedINB, we choose a few IndAEINB with VW histograms (i.e., HistIndAEINB) since they are optimal in terms of overall translation performance and communication cost. Note in this case, autocoders and INB should be considered together as a single translation model $f$. At each batch in the training, we randomly assign a target domain for each sample which is different from the default setup of DIRT (they assign the same target domain for a whole batch in the training). For MNIST, we use 64 as batch size and 0.001 as learning rate. We set the regularization weight of DIRT [10] to be 2. For FashionMNIST, we use 128 as batch size and 0.0001 as learning rate. We set the regularization weight of DIRT to be 10.

**FedAvg** The FedAvg model structure is the same as FedDIRT. We use the uniformly weighted averaging of model updates similar to the original implementation McMahan et al. (2017). We consider three different configurations of FedAvg as baselines: FedAvg with `sync_steps`=1, FedAvg with `sync_steps`=10, and FedAvg with `sync_steps`=100. Learning rate and batch size are the same as those used for corresponding FedDIRT.

## F Additional results

### F.1 FID Score of Results in Section 4

In Figure 4, we include results of FID. One observation is that for FID, the influence of the blurriness caused by applying autoencoders is more significant than the domain shifts. This is unsurprising because FID is based on a modern CNN and CNNs are known to focus on texture or fine-grained

---

[10] The weight before domain invariant feature term.

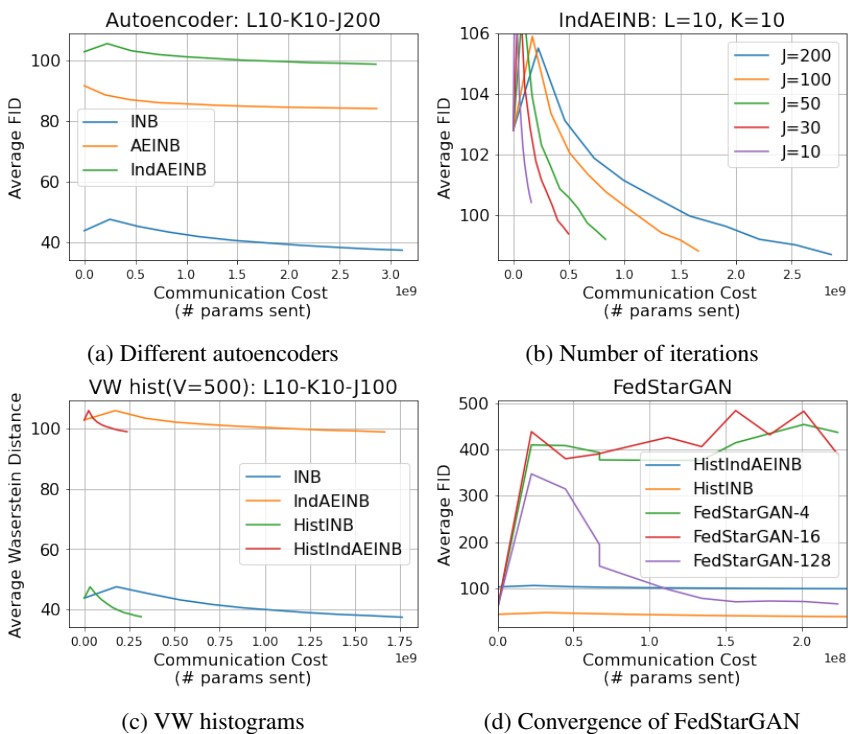

Figure 4: FID (lower is better) as communication cost increases for Rotated MNIST. (a) Different autoencoders used for INB. (b) Different $J$ used for IndAEINB. (c) Using VW histograms for INB. (d) Failure of FedStarGAN. The INB used here is HistIndAEINB-L10-K10-J100 with $V = 500$ and HistINB-L10-K10-J100 with $V = 500$.

details instead of high-level semantics (Geirhos et al., 2019).[11] Because we are proposing a more general approach to federated translation that is not specific to image data, we suggest that the domain-agnostic metric of WD is a better metric for our context rather than the image-specific FID metric. Besides, as shown in Figure 12, AEINB and IndAEINB generate reasonable rotations of the original samples (albeit somewhat blurry because of the AE), and keeping the semantics of the domain translation (i.e., correct rotation) is more important than preserving the fine details for downstream tasks like DG.

### F.2 DIFFERENT SETUPS OF INB WITH ROTATED MNIST

In this section, we include results with more different setups of INB.

**Autoencoder.** As shown in Figure 5, with different setups of INB (different $L, K, J$, whether to use VW histograms and different $r$), we observe that in terms of WD, IndAEINB is better than using a shared autoconder or not using autoencoders.

**Max iterations $J$.** In Figure 6, we include more results with different INB setups to study the influence of decreasing $J$. We can conclude that using a higher $J$ does not lead to a better performance after convergence.

**Number of slices $K$.** As shown in Figure 7, we can observe that as we increase $K$ to 30, the communication cost becomes 3 times higher but the overall performance is not improved. Hence, we use lower $K$ in following experiments to reduce communication cost.

---

[11]Geirhos, Robert, et al. "ImageNet-trained CNNs are biased towards texture; increasing shape bias improves accuracy and robustness." *International Conference on Learning Representations*. 2018.

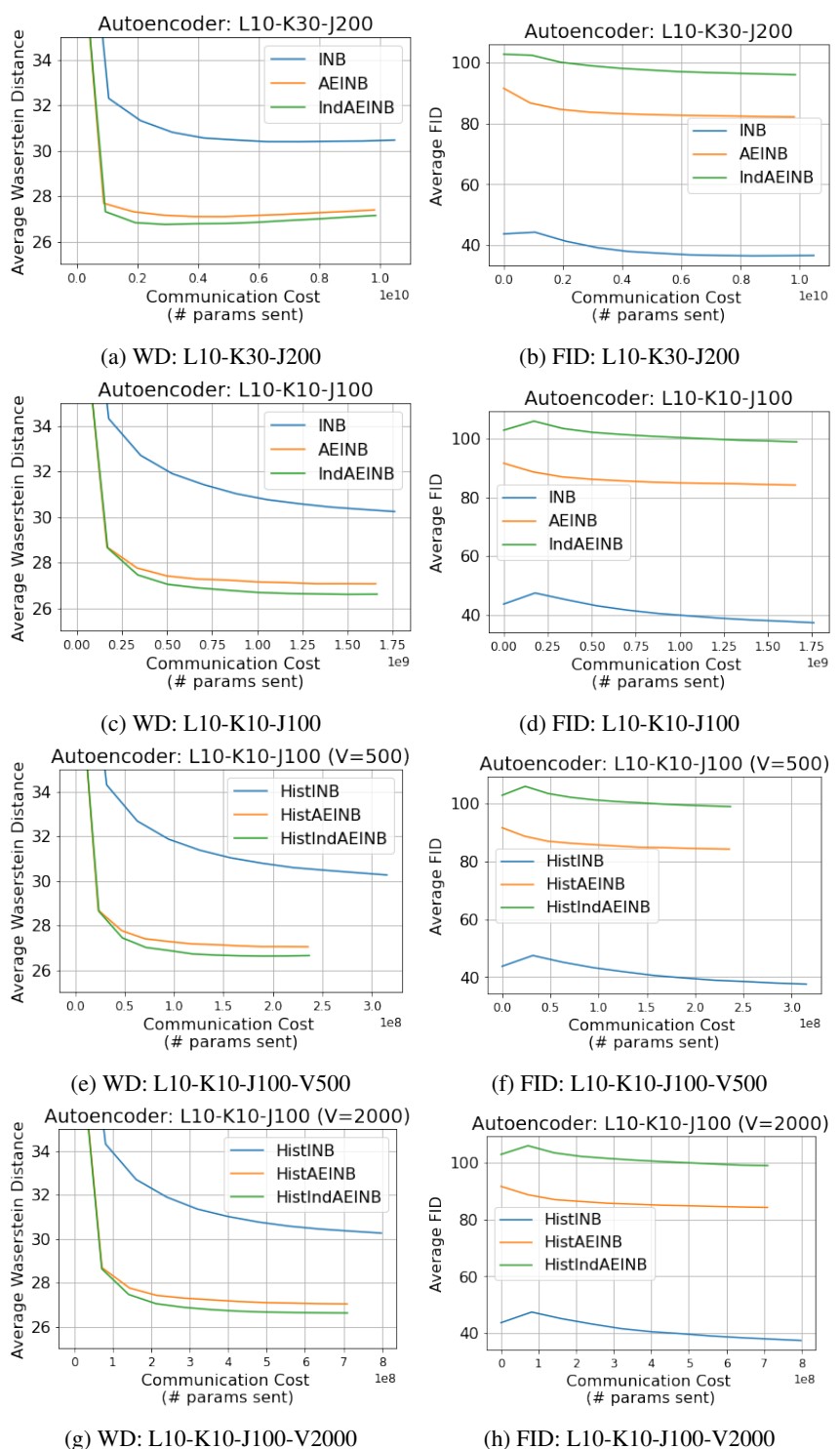

Figure 5: Influence of autoencoders with different setups of INB for Rotated MNIST.

**VW histograms.** As shown in Figure 8, as we decrease $V$, both HistINB and HistIndAEINB can achieve good performance (note the actual number of samples at each client is 10,000).

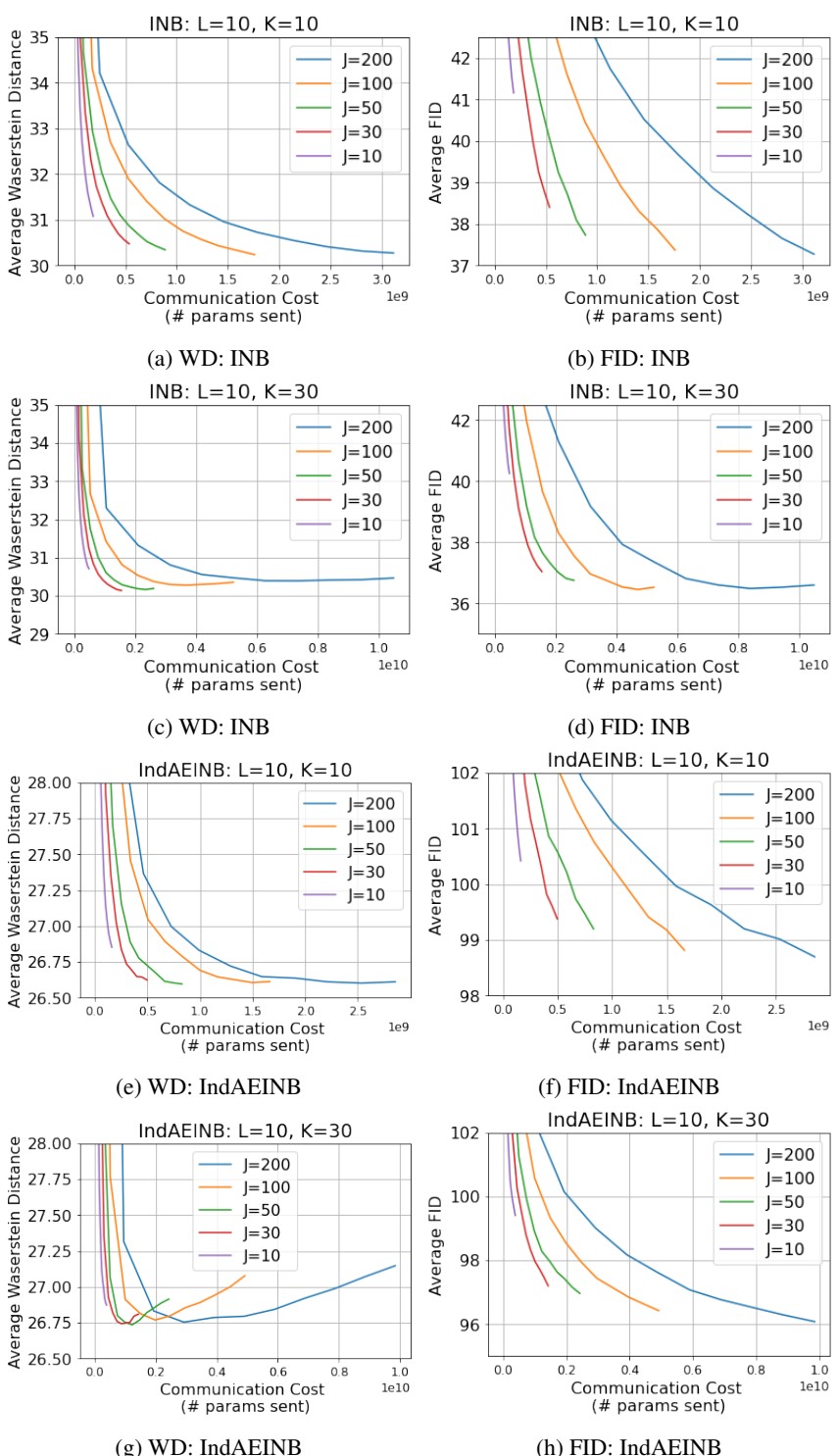

Figure 6: Influence of $J$ for Rotated MNIST.

## F.3 DIFFERENT SETUPS OF INB WITH ROTATED FASHIONMNIST

In this section, we include quantitative federated domain translation results with Rotated FashionM-NIST. The experiment setup here is not exactly the same as that with Rotated MNIST. We skip some

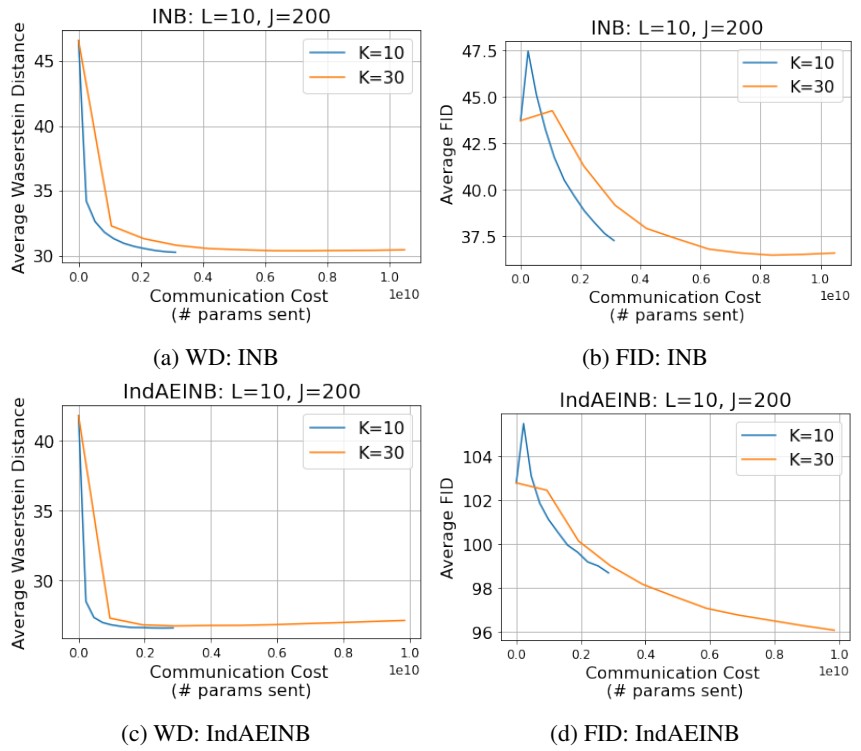

Figure 7: Influence of $K$ for Rotated MNIST.

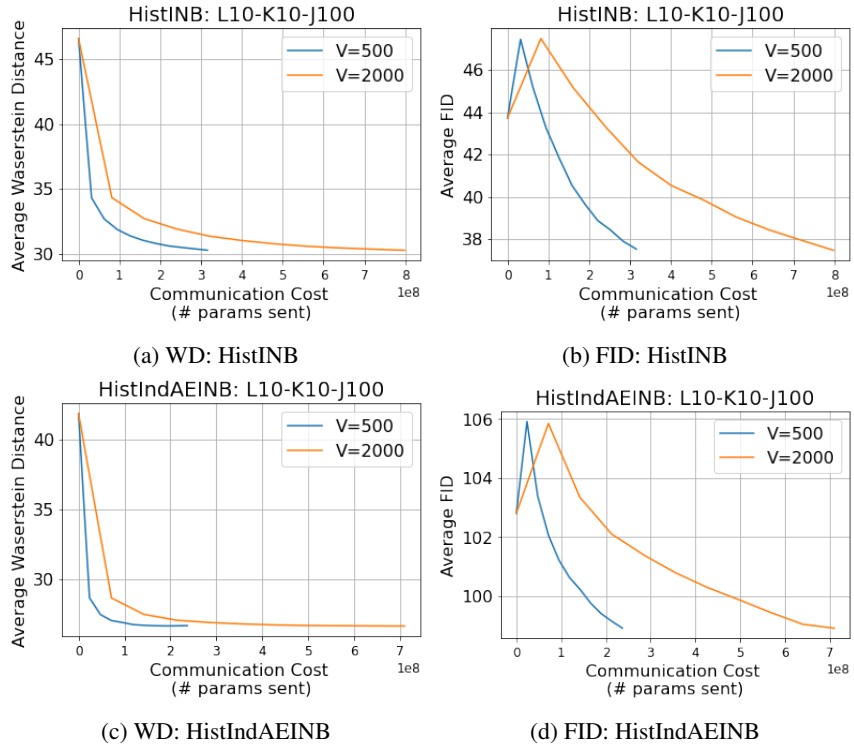

Figure 8: Influence of $V$ for Rotated MNIST.

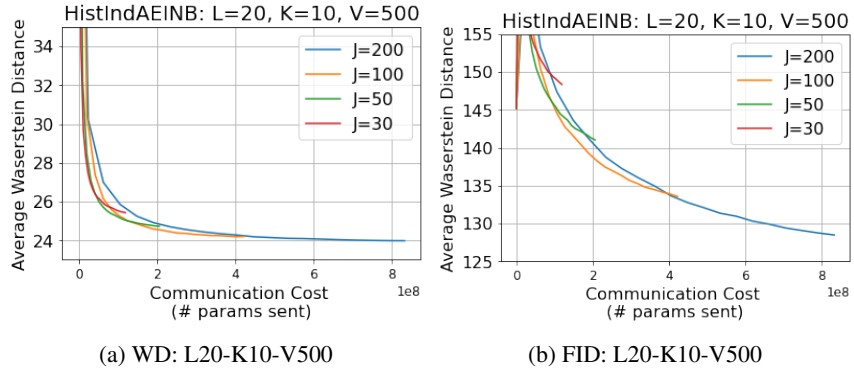

(a) WD: L20-K10-V500        (b) FID: L20-K10-V500

Figure 9: Influence of $J$ for Rotated FashionMNIST.

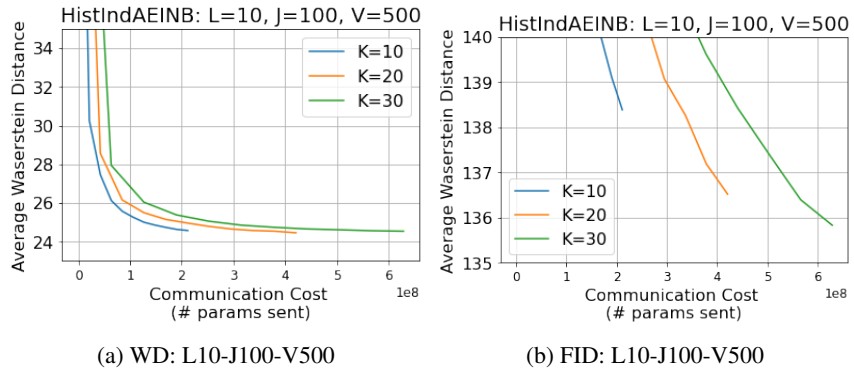

(a) WD: L10-J100-V500        (b) FID: L10-J100-V500

Figure 10: Influence of $K$ for Rotated FashionMNIST.

preliminary tests such as investigation of autoencoders and we test with HistIndAEINB instead of IndAEINB for $J$ and $K$.

**Max iterations $J$.** As shown in Figure 9, we observe that HistIndAEINB with $J = 200$ and $100$ achieve similar performance in terms of WD while $J = 50$ and $30$ seem to perform worse. On one hand, we send histograms instead of all samples (as that in Figure 6 with Rotated MNIST). On other hand, it is inherently a harder task to find the best projection for Rotated FashionMNIST such that more number of iterations is as expected.

**Number of slices $K$.** As shown in Figure 10, as we increase $K$ to 30, the communication cost becomes 3 times higher but the overall performance is not improved in terms of WD.

**Number of histogram bins $V$.** As shown in Figure 11, even if the total number of samples is 10000 and FashionMNIST is a much harder task, the performance of HistIndAEINB does not drop until $V$ is as small as 50.

## F.4 MORE QUALITATIVE TRANSLATION RESULTS

In Figure 12, we include more qualitative results with MNIST. We observe that in comparison to INB, IndAEINB and HistIndAEINB generate more smooth samples. Moreover, samples translated by IndAEINB and HistIndAEINB are very similar to each other. In Figure 13, we include qualitative results with FashionMNIST.

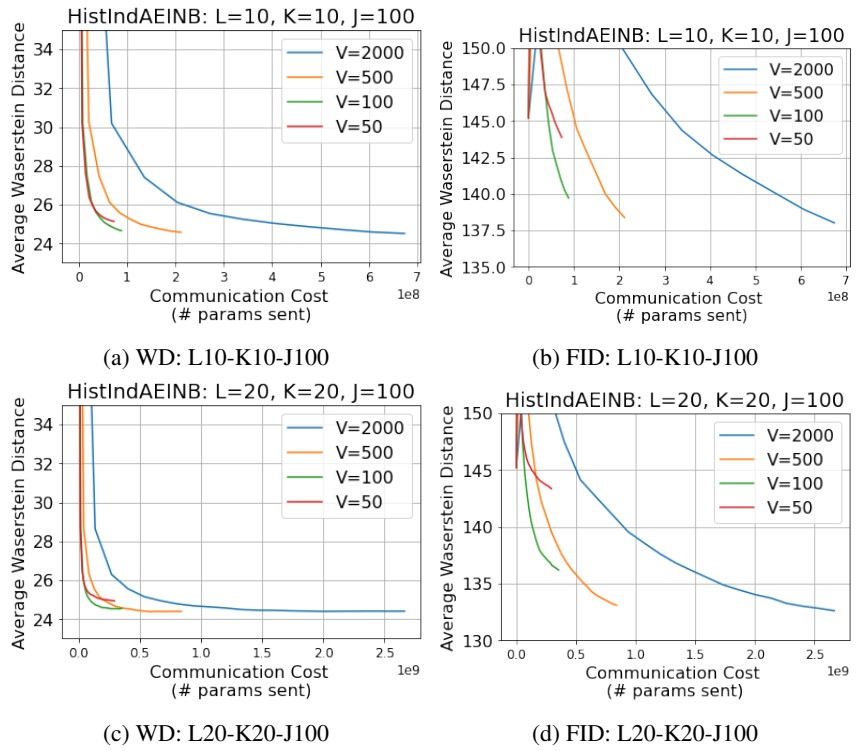

Figure 11: Influence of $V$ for Rotated FashionMNIST.

## F.5 MORE RESULTS OF FEDERATED DOMAIN GENERALIZATION

In Table 3 and Table 4, we include results with more INB setups. We can observe that HistIndAEINB achieves similar performance, and they are all better than FedAvg. In Figure 14, we plot the test accuracy of FedDIRT and FedAvg to demonstrate that the training already converges when we record the test accuracy.

## F.6 INVESTIGATION OF THE OPTIMIZATION OF FEDINB

In this section, to further compare the optimization of FedINB and original INB, we investigate the optimization of the multi-max-K-SW subproblem throughout the full optimization process with respect to the AE and VW histograms improvement. Note that the convergence of the whole algortihm has been investigated in Figure 2a, Figure 2c, Figure 5, and Figure 8, which indicates that autoencoders lead to a better alignment and VW histograms do not hurt the performance unless $V$ is really small. So we focus on their impact on multi-max-K-SW here.

As a reminder, the objective of multi-max-K-SW is $\frac{1}{MKn}\sum_{m=1}^{M}\sum_{k=1}^{K}\sum_{i=1}^{n}|(\boldsymbol{\theta}_k^T\boldsymbol{x}_m)_{[i]} - \boldsymbol{y}_{[i],k}|^2$. In Figure 15, we track the change of maximum K-sliced Wasserstein Divergence through the training of a 10-layer ($L = 10$) FedINB for different digits where we constrain the maximum number of iterations to be 100 ($J = 100$). We can observe that using VW histograms barely leads to any difference in the optimization: the point they converge to and the speed they converge at are very similar. For IndAEINB, we cannot directly compare the numbers as they are essentially computed in difference spaces. But we can observe that IndAEINB actually converges faster especially in the last layers, and the max-K-SW is much closer to 0 in the last few layers. This could result from that we are running multi-max-K-SW in a space in much lower dimension and the task becomes easier. Besides, we observe that in most cases, multi-max-K-SW keeps running till it reaches the maximum number of iterations even if there is barely any improvement. This also explains why we could reduce $J$ without hurting the performance (as explored in Section 4.1). In Table 5, we also track the total number of iterations with different setups. We can observe that INB and HistINB requires similar iterations while AEINB takes relatively less iterations.

Table 3: Classification accuracy at unseen domains for Rotated MNIST. For each target domain, we try to aggregate model after each 1, 10, 100 mini-batches. The mean and standard deviation are taken over 4 runs. For FedDIRT, we use HistIndAEINB with $J = 100$ and $V = 500$ (with different $L$ and $K$)

| Model | **75** (1-batch) | **75** (10-batch) | **75** (100-batch) | **90** (1-batch) | **90** (10-batch) | **90** (100-batch) |
|---|---|---|---|---|---|---|
| FedDIRT(L10-K20) | **92.4 ± 0.3** | 91.7 ± 1.2 | 90.2 ± 1.0 | 70.1 ± 0.3 | 68.5 ± 3.8 | 65.7 ± 2.9 |
| FedDIRT(L20-K10) | 92.2 ± 0.3 | 89.3 ± 4.2 | **91.9 ± 0.9** | 69.8 ± 0.7 | 70.4 ± 2.2 | **69.9 ± 0.3** |
| FedDIRT(L10-K10) | **92.4 ± 0.4** | **92.2 ± 0.7** | 90.9 ± 0.8 | **70.5 ± 0.5** | 70.8 ± 0.8 | 65.9 ± 4.7 |
| FedDIRT(L20-K20) | 92.2 ± 1.3 | 90.8 ± 1.2 | 91.4 ± 0.6 | 69.4 ± 0.8 | **71.1 ± 1.3** | 69.8 ± 1.5 |
| FedAvg | 85.2 ± 0.7 | 85.1 ± 0.5 | 80.1 ± 2.3 | 63.8 ± 2.1 | 63.6 ± 0.8 | 55.6 ± 2.0 |

Table 4: Classification accuracy at unseen domains for Rotated FashionMNIST. For each target domain, we try to aggregate model after each 1, 10, 100 mini-batches. The mean and standard deviation are taken over 4 runs. For FedDIRT, we use HistIndAEINB with $J = 100$ and $V = 500$ (with different $L$ and $K$)

| Model | **75** (1-batch) | **75** (10-batch) | **75** (100-batch) | **90** (1-batch) | **90** (10-batch) | **90** (100-batch) |
|---|---|---|---|---|---|---|
| FedDIRT(L10-K20) | **65.8 ± 1.3** | 64.1 ± 0.9 | 62.9 ± 1.4 | 18.4 ± 0.5 | 18.3 ± 0.6 | **19.0 ± 0.4** |
| FedDIRT(L20-K10) | **65.8 ± 1.3** | 63.6 ± 0.9 | 63.0 ± 1.0 | 18.2 ± 0.9 | 18.1 ± 0.4 | 18.8 ± 0.6 |
| FedDIRT(L10-K10) | 65.1 ± 1.8 | 63.5 ± 0.6 | 62.6 ± 0.6 | 17.9 ± 0.5 | 18.3 ± 0.3 | 18.8 ± 0.6 |
| FedDIRT(L20-K20) | 65.5 ± 1.8 | **64.5 ± 1.5** | **63.5 ± 1.3** | **19.0 ± 0.3** | **18.5 ± 1.0** | 18.9 ± 0.4 |
| FedAvg | 51.9 ± 2.9 | 50.4 ± 1.2 | 40.4 ± 4.1 | 13.7 ± 1.6 | 14.6 ± 1.8 | 13.1 ± 0.9 |

Table 5: Number of iterations of multi-max-K-SW with $L = 10$, $K = 10$, $J = 100$ for RotatedMNIST. The numbers are averaged over 4 runs. Note that the extra steps ($L \times J = 10 \times 100 = 1000$) come from the backtracking line search as explained in Appendix C.3.

| | **INB** | **HistINB** | **IndAEINB** |
|---|---|---|---|
| **Digit 0** | 1,605.25 | 1,605.00 | 1,586.50 |
| **Digit 1** | 1,609.00 | 1,612.75 | 1,580.50 |
| **Digit 2** | 1,601.75 | 1,603.50 | 1,584.75 |
| **Digit 3** | 1,599.50 | 1,600.00 | 1,583.25 |
| **Digit 4** | 1,597.75 | 1,594.50 | 1,581.25 |
| **Digit 5** | 1,605.00 | 1,605.25 | 1,587.75 |
| **Digit 6** | 1,601.00 | 1,601.00 | 1,582.75 |
| **Digit 7** | 1,600.75 | 1,601.25 | 1,581.25 |
| **Digit 8** | 1,601.50 | 1,600.75 | 1,583.25 |
| **Digit 9** | 1,598.75 | 1,599.75 | 1,580.00 |
| **Average** | 1,602.03 | 1,602.38 | 1,583.13 |

In Figure 16, we increase the maximum number of iterations to be 300 and observe similar results. In Table 6, we observe that total number of iterations of HistINB is slightly larger than INB , but in practice, we don't actually need this many iterations.

From Figure 15, we also observe that in the last few layers, the maximum K-sliced Wasserstein Divergence found by the algorithm is very small which is consistent with the observation of convergence of FedINB in Figure 2c.

Table 6: Number of iterations of multi-max-K-SW with $L = 10, K = 10, J = 300$ for RotatedM-NIST. The numbers are averaged over 4 runs.

|  | INB | HistINB | IndAEINB |
|---|---|---|---|
| **Digit 0** | 4,767.67 | 4,800.67 | 4,756.50 |
| **Digit 1** | 4,780.00 | 4,782.67 | 4,756.50 |
| **Digit 2** | 4,778.00 | 4,777.67 | 4,756.00 |
| **Digit 3** | 4,779.00 | 4,779.33 | 4,753.50 |
| **Digit 4** | 4,778.67 | 4,781.67 | 4,754.50 |
| **Digit 5** | 4,778.67 | 4,784.33 | 4,758.50 |
| **Digit 6** | 4,743.33 | 4,788.00 | 4,753.50 |
| **Digit 7** | 4,776.67 | 4,786.00 | 4,754.50 |
| **Digit 8** | 4,779.00 | 4,778.67 | 4,754.50 |
| **Digit 9** | 4,781.67 | 4,789.00 | 4,754.00 |
| **Average** | 4,774.27 | 4,784.80 | 4,755.20 |

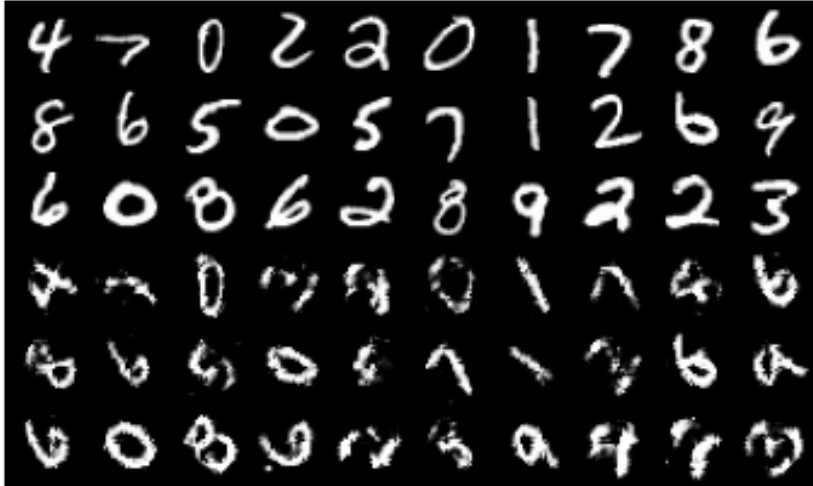

(a) INB: L=10, K=10, J=200

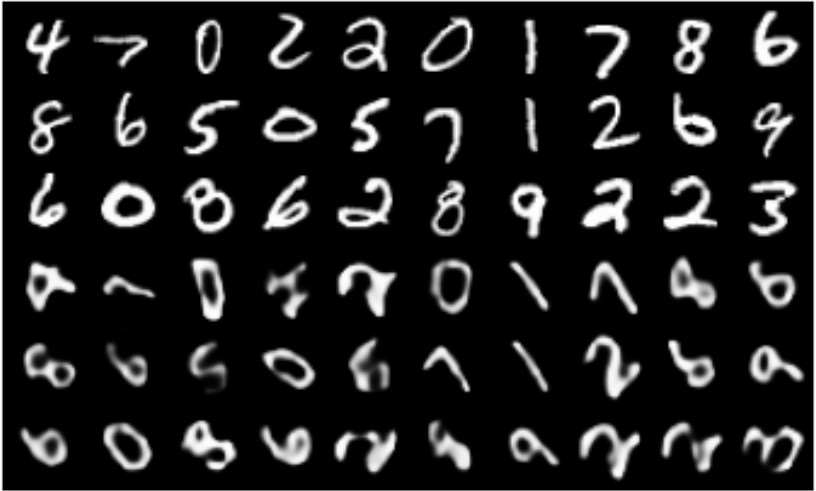

(b) IndAEINB: L=10, K=10, J=200

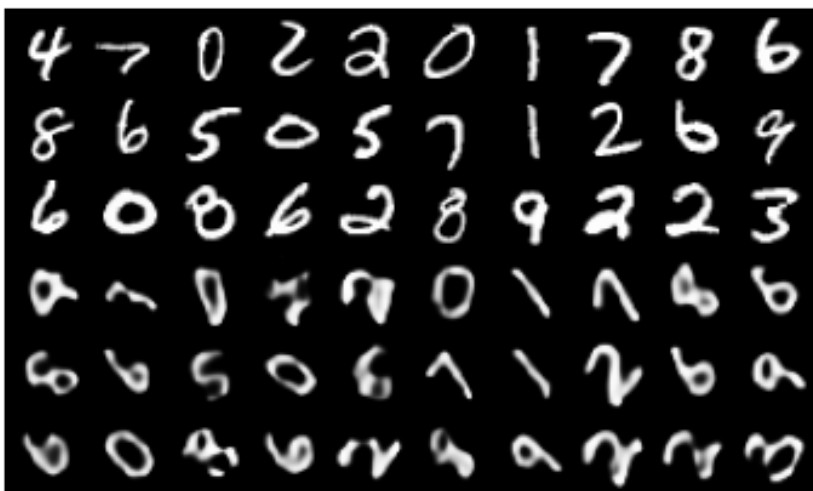

(c) HistIndAEINB: L=10, K=10, J=200, V=500

Figure 12: More qualitative results for Rotated MNIST. The first three rows are original samples from domain 0. The last three rows are translated samples (from domain 0 to 4).

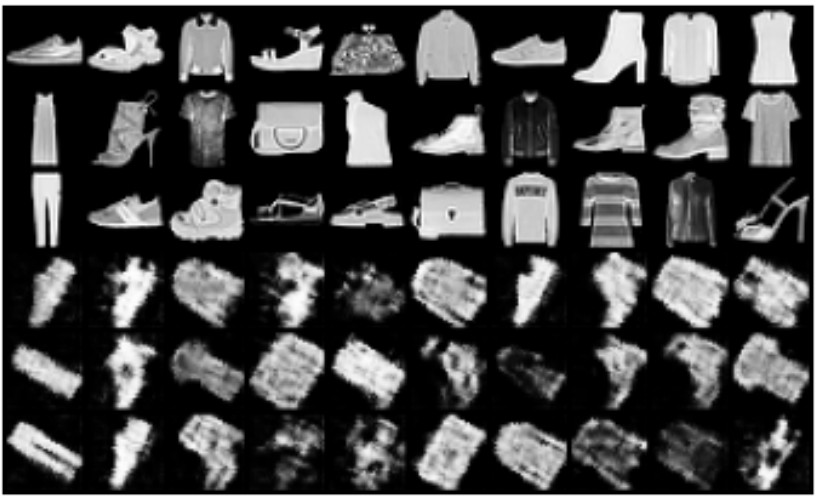

(a) HistIndAEINB: L=10, K=10, J=100, V=500

Figure 13: Qualitative results for Rotated FashionMNIST. The first three rows are original samples from domain 0. The last three rows are translated samples (from domain 0 to 4).

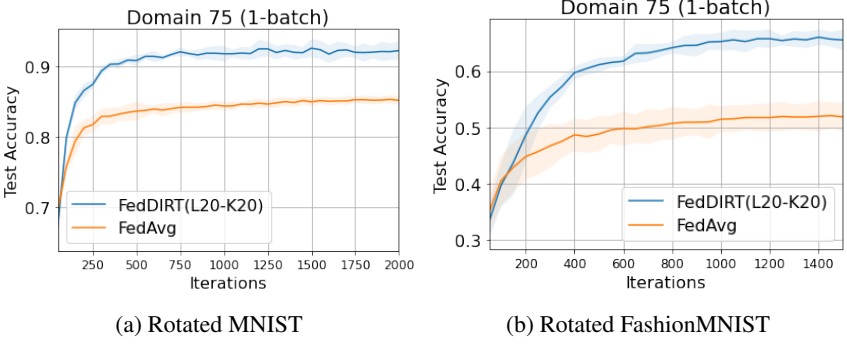

(a) Rotated MNIST            (b) Rotated FashionMNIST

Figure 14: Test accuracy in the federated domain generalization experiments.

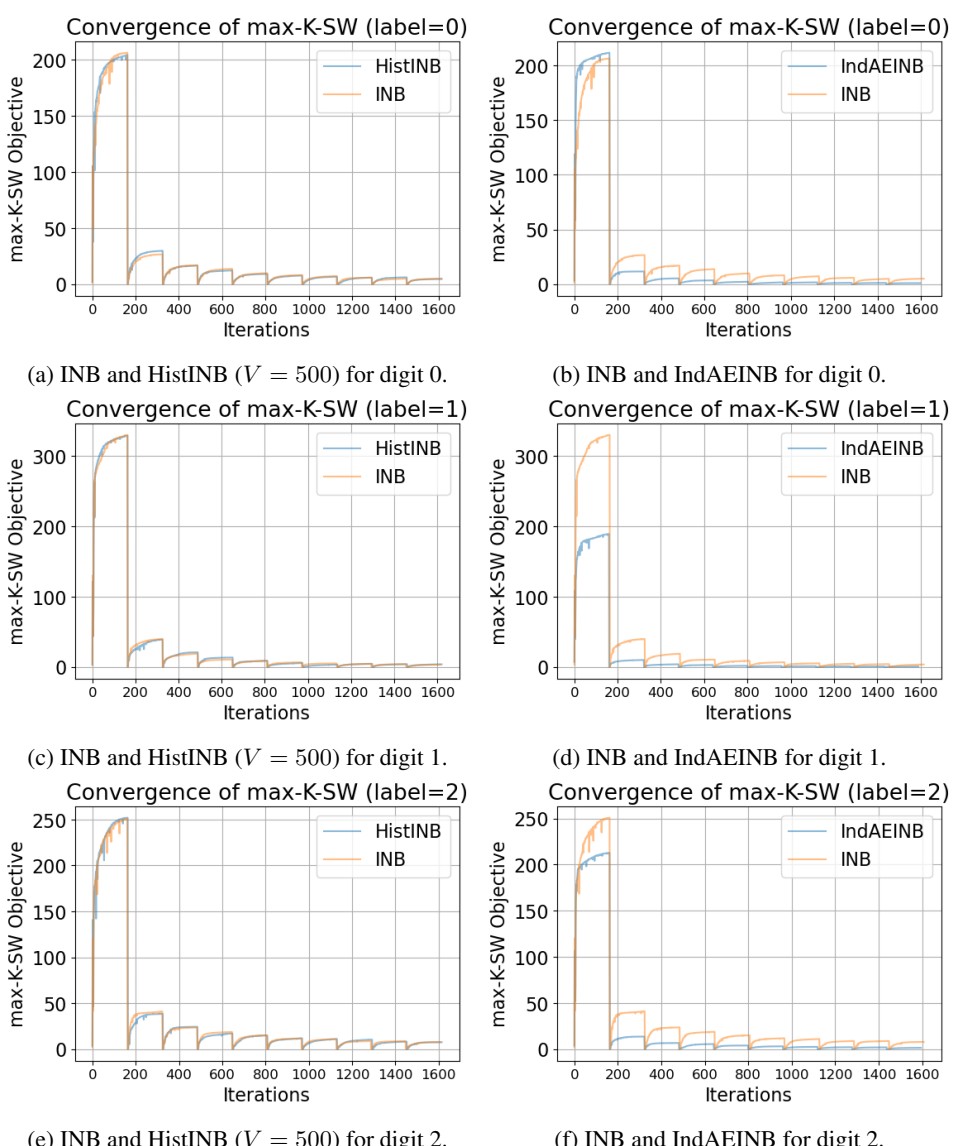

(a) INB and HistINB ($V = 500$) for digit 0.

(b) INB and IndAEINB for digit 0.

(c) INB and HistINB ($V = 500$) for digit 1.

(d) INB and IndAEINB for digit 1.

(e) INB and HistINB ($V = 500$) for digit 2.

(f) INB and IndAEINB for digit 2.

Figure 15: Convergence of multi-max-K-SW with $L = 10, K = 10, J = 100$ for RotatedMNIST. The curves describe the change of loss through 10 layers of FedINB with different setups. The sudden change of loss is because we apply 1D-Barycenter there to align the projected distribution. We want to note that the value of objective for INB and IndAEINB is not directly comparable because for IndAEINB we are applying FedINB in a space in much smaller dimension.

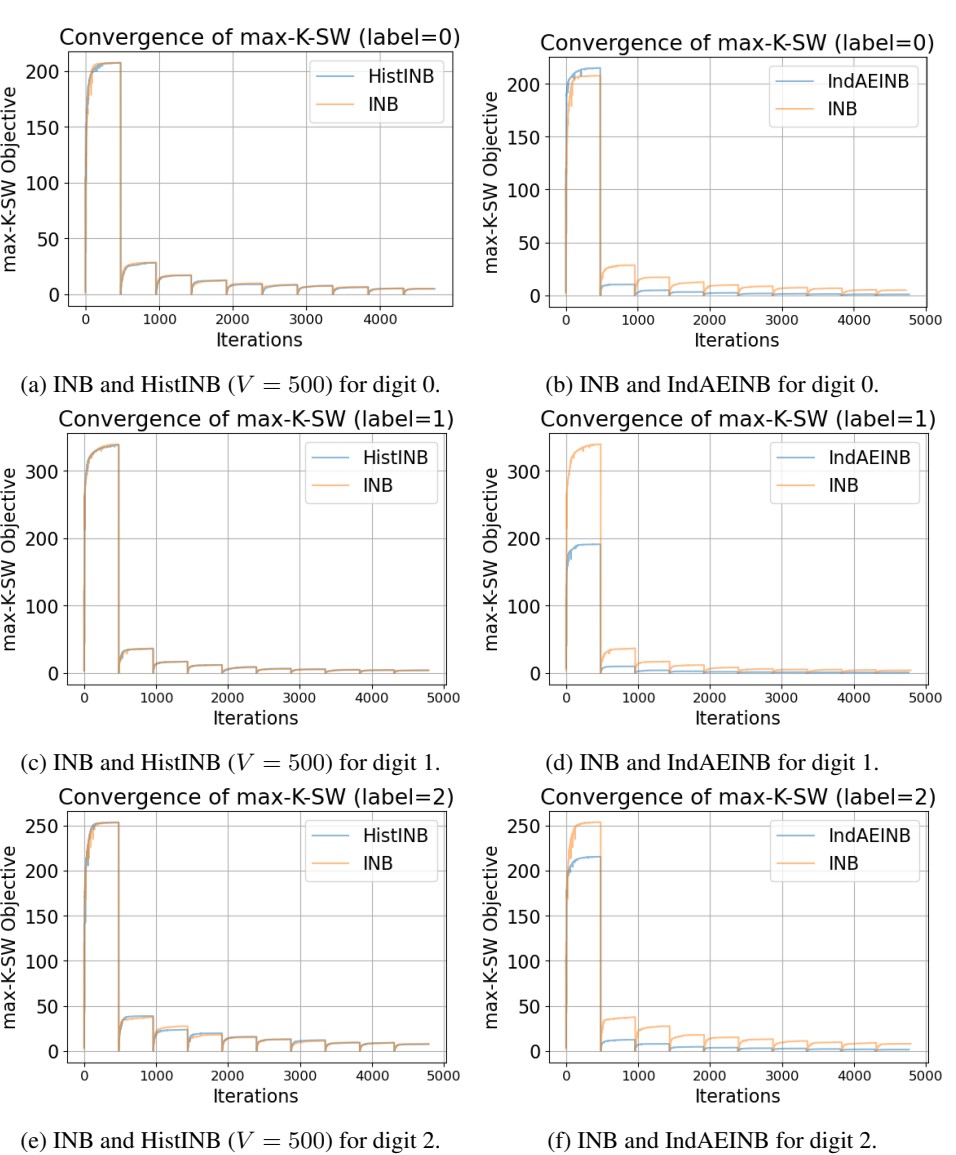

(a) INB and HistINB ($V = 500$) for digit 0.

(b) INB and IndAEINB for digit 0.

(c) INB and HistINB ($V = 500$) for digit 1.

(d) INB and IndAEINB for digit 1.

(e) INB and HistINB ($V = 500$) for digit 2.

(f) INB and IndAEINB for digit 2.

Figure 16: Convergence of multi-max-K-SW with $L = 10, K = 10, J = 300$ for RotatedMNIST.

