# OpenReview forum: "Efficient Federated Domain Translation "
_ICLR.cc/2023/Conference — ICLR 2023 poster_

### Official Review · Reviewer_zb7q · 2022-10-20

**Confidence:** 2
**Correctness:** 3
**Technical Novelty And Significance:** 3
**Empirical Novelty And Significance:** Not applicable
**Recommendation:** 8

**Clarity, Quality, Novelty And Reproducibility:**

The paper is written very well and concise. The extensive abstract should contain everything necessary to reproduce their results.

**Strength And Weaknesses:**

I am no expert in federated learning, so I cannot comment on how novel the contributions are and how relevant this work is to practical federated learning. But on a high level, the paper seems to be well motivated. I could easily follow the overall story. I did not check 100% of the formalisms, but I could not find any vagueness/clarity issues in the ones I did check. The claims regarding communication efficiency are clearly demonstrated in Fig. 2.

Tables 1 and 2 are also clearly demonstrating the benefits regarding domain generalization, but I'm wondering why they do not include a baseline from Liu et al. (2021b) (see Sec. 3). Also, if the advantage over Liu et al. (2021b) is that this work is amenable to other data modalities such as text, it would be nice to evaluate on a text-based task.

**Summary Of The Paper:**

The paper starts with pointing out that the prevalent perspective regarding modeling heterogeneous FL clients by looking at the label distribution is limiting, since, in practice, the data distribution may differ between clients, too. The authors propose to mitigate this concern using a variant of an iterative domain translation model that they adapted to the FL scenario. They evaluate on a one-domain-per-client setting using Rotated MNIST and FashionMNIST.

**Summary Of The Review:**

To a reviewer without a background in federated learning, this looks like a strong submission.

---

> ### Author Response · Authors · 2022-11-09
> **Response to Reviewer zb7q**
>
> We thank the reviewer for the positive review and valuable comments. We respond to your individual comments below and have changed the paper accordingly (also highlighted in blue).
>
>
> **Why not include a baseline from Liu et al. (2021b)?**
>
> There are two main reasons why we do not include ELCFS from Liu et al. (2021b). First, ELCFS requires sharing the amplitude spectrum of the original data, which could lead to data leakage and privacy concerns that are a major emphasis in federated learning. We have a more detailed discussion on this in Appendix D. Second, ELCFS is designed specifically for image segmentation tasks. It is unclear how one could adjust their Boundary-oriented Episodic Learning (BEL) methodology to accomodate general classification tasks. We have adjusted our phrasing in Section 3 of the revised manuscript to clarify these points.
>
> **If the advantage over Liu et al. (2021b) is that this work is amenable to other data modalities such as text**
>
> We apologize for the confusion here due to unclear phrasing in Section 3. The modality is not the main reason why we do not include ELCFS as a baseline; rather, it is the two points above. However, it is also true that our method is more general and could be applied to different data modalities in theory. For example, with text data, one could embed the data in a continuous space using a pretrained model and naturally apply our translation model. This would be beneficial to federated tasks which require information of other domains such as federated domain generalization.
>
> Please let us know if you have any further comments. We look forward to hearing from you again soon.

---

> ### Author Response · Authors · 2022-11-16
> **Follow-up**
>
> Hi Reviewer zb7q, thank you for your helpful feedback. Have we adequately addressed your questions and concerns?

---

### Official Review · Reviewer_CNpn · 2022-10-26

**Confidence:** 2
**Correctness:** 3
**Technical Novelty And Significance:** 3
**Empirical Novelty And Significance:** 2
**Recommendation:** 6

**Clarity, Quality, Novelty And Reproducibility:**

The paper is basically clear, with solid experiments and demonstration to previous arguments. Its idea is somewhat incremental, but practical improvement are presented to reduce the communication cost of the original method.

**Strength And Weaknesses:**

Strength:

1. Novelty: The authors propose to apply INB in FL setting for the problem where different clients have their own class distribution. In addition, the authors propose several FL-motivated improvements to INB, including the use of variable-bin-width histograms and autoencoder.
2. Performance: Empirical results on Rotated MNIST and FashionMNIST show that the proposed method outperforms FL versions of standard translation models under practical settings of limited client-server communication capability.

Weakness and Suggestions:

While it does propose practical improvement, the work seems incremental in applying INB in federated learning.

The key point in INB is to learn the transformation which maps each domain distribution to a shared latent distribution. While it could be reasonable or practical for some domains, there could be cases where the mapping is difficult to learn. This may need to be discussed as potential weaknesses.

I am wondering how much the difference is between a INB and a FedINB, where there be a balance/trade-off between communication and the performance?


**Summary Of The Paper:**

This paper focuses on a challenging setting in federated learning (FL) where class conditional distribution varies amongst clients. To address this issue, the authors develop a federated domain translation methodology based on an advanced iterative translation model, namely Iterative Naive Barycenter(INB).
INB is more viable to the FT setting than standard translation methods, e.g. StarGAN. Furthermore, the authors propose several FL-motivated improvements to INB, which significantly reduce communication costs. Empirical results show that the proposed federated translation model not only performs better than FL versions of baseline translation model, but also improves robustness

**Summary Of The Review:**

This is a solid paper which applies INB in a Federated Learning setting with practical improvement to reduce the communication cost.

---

> ### Author Response · Authors · 2022-11-09
> **Response to Reviewer CNpn (1 of 2)**
>
> We thank the reviewer for your positive review. We respond to the questions and suggestions individually below. We have also modified the paper accordingly (highlighted in blue).
>
>
> **While it does propose practical improvement, the work seems incremental in applying INB in federated learning.**
>
> We point out the novelty of our work in the response to all reviewers above. We repeat those points here for convenience,
>
> >Our central contribution is **developing methods that address the novel problem of** ***FL translation*** that has not been considered in the literature before. Domain translation in FL is a very difficult task, particularly for the realistic use case where each client represents a domain, as we demonstrate empirically in the paper. **Typical translation models cannot train under communication and privacy constraints.** This novel problem setting motivates our choice of algorithm (i.e., INB over standard translation models), our theoretical developments (e.g., showing that the Fed algorithm is equivalent to the non-Fed algorithm), and our engineering innovations (e.g., variable-bin-width histograms, sorting before transmission for privacy, and pseudo-empirical barycenter samples for communication and privacy).
> >
>
> >More specifically, first, we prove that federated multi-max-K-SW is equivalent to the original multi-max-K-SW. This is an **important finding for enabling usage of INB in the federated setting**: INB backpropagates through $\boldsymbol{y}$ when computing $\nabla_{\boldsymbol{\theta}} \boldsymbol{d}_m$, which requires keeping track of how $\boldsymbol{y}$ is computed from each $\boldsymbol{x}_m$. This could lead to both communication and privacy concerns. Instead, we show that treating $\boldsymbol{y}$ as a constant does not lead to any change in the gradient computation, which alleviates these concerns. Second, we propose to use **VW histograms** instead of directly sending samples over the network. This significantly improves privacy and resource efficiency---again, an innovation that is novel compared to INB and is motivated by FL concerns. We empirically show that the performance does not decrease with this FL-motivated innovation. Third, we **empirically demonstrate that our FedINB approach performs significantly better** than FL versions of standard translation models under settings with limited client-server communication capability, which is of critical practical importance in FL. Fourth, we **empirically demonstrate the feasibility of FedINB for federated domain generalization**, which is a challenging practical problem with few existing solutions. We also show that our federated DG method provides substantial improvements in robustness to an increasing synchronization period, allowing reductions in communication overhead, which is again important in FL settings.
> >
>
>
> **I am wondering how much the difference is between an INB and a FedINB  where there is a balance/trade-off between communication and the performance?**
>
> We discuss the novelty of FedINB in comparison to INB above. In theory, the core algorithm of INB and FedINB should lead to the same result as we prove in Remark 1. In practice, we use VW histograms and autoencoders to improve efficiency and actually observe improved performance in certain cases. As can be seen in Figure 2(c), the VW histogram improvement does not lead to drop in performance while significantly reducing the communication cost, and adding autoencoders leads to significant improvement in both communication and performance. As shown in Figure 8 and Figure 11, there is only a drop in performance when the maximum number of histogram bins $V$ is very small.

---

> > ### Author Response · Authors · 2022-11-09
> > **Response to Reviewer CNpn (2 of 2)**
> >
> > **While it could be reasonable or practical for some domains, there could be cases where the mapping is difficult to learn. This may need to be discussed as potential weaknesses.**
> >
> > We agree there could be cases where the mapping is hard to find. For example, FedINB is a greedy algorithm. On one hand, this leads to benefits in computation and communication. On the other hand, this does not guarantee we find the global optimum of the alignment problem. From a theoretical perspective, as pointed out in Appendix C.2 of [Zhou et al., 2022], the best we can guarantee even for the centralized case is that at each iteration, given the current $\boldsymbol{\theta}$, the local alignment map $t_m$ is optimal. Thus, in optimizing $\boldsymbol{\theta}$, especially considering the addition of VW histogram and reducing the maximum number of iterations $J$ due to communication concerns, the empirical maximization of multi-max-K-SW may not find the true maximum. But, as we empirically show in the experiment section, neither leads to drop in performance unless $J$ or the maximum number of histogram bins $V$ is really small. To further improve this fundamentally, we could investigate more on the optimization on the manifold of orthonormal matrices.
> >
> > We have discussed some limitations of INB in Section 5 and Appendix C.4. To address this comment, we have added the argument above in Appendix C.4 of the revised manuscript.
> >
> > Please let us know if you have any further comments. We look forward to hearing from you again soon.

---

> > > ### Comment · Reviewer_CNpn · 2022-11-16
> > > **about the learning process**
> > >
> > > Indeed, the optimization itself has not guarantee of a global optimal. My question is more about how the techniques in FedINB affects the learning process. For example, will it (FebINB) be more likely to stuck in local optimal? Or, will it affect the process of convergence?

---

> > > > ### Author Response · Authors · 2022-11-17
> > > > **Investigation of the learning process**
> > > >
> > > > Thank you for clarifying the question. To address your concern, we add some extra investigations in Appendix F.6 to check if the convergence of **the inner problem** (i.e., multi-max-K-SW) is fundamentally changed by our VW histograms and AE innovations (we showed that they do not hurt performance of the overall algorithm in Figure 2a, Figure 2c, Figure 5, and Figure 8). From our investigations, VW histograms do not seem to hurt the convergence of multi-max-K-SW and autoencoders lead to a faster convergence of multi-max-K-SW. More details can be found in the revised manuscript in Appendix F.6. Does this answer your question?

---

> > > > > ### Comment · Reviewer_CNpn · 2022-11-30
> > > > > **About the organization of the paper**
> > > > >
> > > > > Dear authors,
> > > > >
> > > > > Thanks for your response and follow-up experiments. I have no concern about the technical part now. However, I noticed that quite a lot of experiments are done and included in the appendix, I think it will be better to reconsider the structure/organization of the whole paper, so it will be more friendly for first-readers.
> > > > >
> > > > > Good luck with your work.

---

> > > > > > ### Author Response · Authors · 2022-11-30
> > > > > > **Response to Reviewer CNpn**
> > > > > >
> > > > > > Dear Reviewer CNpn,
> > > > > >
> > > > > > Thank you for your response. We are happy to hear that you have no more concerns on the technical part. Per your suggestion, we can certainly move some more of the experiments to the main text in the final version of the manuscript (unfortunately, at this point in the review process, ICLR does not permit us to upload a new version here). In light of this, could we ask whether you have considered increasing your score?

---

> ### Author Response · Authors · 2022-11-16
> **Follow-up**
>
> Hi Reviewer CNpn, thank you for your helpful feedback. Have we adequately addressed your questions and concerns?

---

### Official Review · Reviewer_uckp · 2022-10-27

**Confidence:** 2
**Correctness:** 3
**Technical Novelty And Significance:** 3
**Empirical Novelty And Significance:** 3
**Recommendation:** 6

**Clarity, Quality, Novelty And Reproducibility:**

Some writings in the paper are not clear to me, to name a few:

- Most existing non-IID FL works assume that the client distributions only exhibit class imbalance, i.e., the marginal class distributions are different (pm(y)  is different to pm′ (y)), but the class-conditional distributions are equal (pm(x|y) = pm′ (x|y)). In contrast, we focus on the case where the class-conditional distributions differ across clients, i.e., pm(x|y)  is differrent to pm′ (x|y). -> Are the marginal class distributions also different in this study? (i.e. pm(y)  is different to pm′ (y))

- algorithm 1: d is never explained ({Server} Randomly initialize θ ∈ Rd)

- algorithm 1: [i] should be j to my best understanding

- algorithm 1: I don't understand why the barycenter is  y[i],k instead of y,k

- However, we can show that treating y[i],k as a constant will actually return the same gradient value as if it had been treated as a function of θ -> I found this part is both interesting and important, thus should be elaborated more in the paper.



**Strength And Weaknesses:**

* Strength:

The proposed framework sounds to me. I also found the experiments solid, showing the proposed model is more efficient in terms of both communication and computational cost than FedStarGAN. Experiments also show that practical improvements proposed in the paper seem helpful to reduce communication costs.

* Con:

I found some parts of the writing are not clear. However, I also must say that I am not the particularly familiar with the topic, so it also contributes to some difficulty in following the paper to me (see Clarity section).

**Summary Of The Paper:**

The paper proposes a federated domain translation approach that can mitigate conditional shift issues in federated learning tasks. The proposed translation model is empirically shown that it performs better than the state of the art FedStarGAN. The paper also shows that by combining FedINB with DIRT, regularization from domain translation model significantly improves the model’s ability to generalize to unseen domains. I am not particularly familiar with the topic to have indepth comments. However I think the work is solid and should be accepted to ICLR.

**Summary Of The Review:**

An well-studied federated domain translation approach that can mitigate conditional shift issues in federated learning tasks.

---

> ### Author Response · Authors · 2022-11-09
> **Response to Reviewer uckp**
>
> Thank you for your positive review. Note that we have further emphasized our novelty in the response to all reviewers. We respond to the additional questions and suggestions you made individually below, with pointers to locations in the manuscript where the corresponding changes have been made (also highlighted in blue).
>
> **Are the marginal class distributions also different in this study? (i.e. $p_m(y)$ is different to $p_{m′} (y))$?”**
>
> In this work, we are interested in the conditional shift case, and we train FedINB to align the class-conditional distribution, i.e., $p_m(x \mid y)$ and $p_{m’}(x \mid y)$. In our experiments for Rotated MNIST and FashionMNIST, the marginal class distributions are the same. For brevity in the main manuscript, these details can be found in Appendix E.1.
>
> **Algorithm 1: d is never explained ({Server} Randomly initialize $θ ∈ R^{d \times K}$)**
>
> $d$ is the dimension of the data in the original space in contrast with $K$ which is the dimension of the data after projection. For MNIST and FashionMNIST, $d=784$. We have added this clarification in the main paper after Equation (2).
>
> **Algorithm 1: [i] should be j to my best understanding**
>
> We apologize for the confusion resulting from not clearly defining the $[i]$ in the paper. Here $j$ represents the iteration index, which does not explicitly show up in the algorithm at each iteration. $[i]$ instead represents the sorted index (in ascending order) along the axis of the number of samples. For example, $\boldsymbol{\theta}_k^T \boldsymbol{x}_m$ is a vector of dimension $1\times n$ and $(\boldsymbol{\theta}^T _k \boldsymbol{x} _m) _{[i]}$ represents the i-th smallest value of the vector. If $x=[2,3,1]$, then $x _{[1]} = x_3=1$, $x _{[2]} = x_1=2$, and $x _{[3]} = x_2=3$. We have added this clarification in the main paper after Equation (2).
>
> **Algorithm 1: why is the barycenter y[i],k instead of y,k**
>
> The empirical barycenter is expressed as $\boldsymbol{y} _{[i], k} =\frac{1}{M} \sum _{m=1}^M \left(\boldsymbol{\theta}^T_k \boldsymbol{x} _m \right) _{[i]}$. $[i]$ is used here to be consistent with the right-hand side of the equation, indicating the sorted index of the vector. It is also important to note here that $\boldsymbol{y}_k$ is naturally sorted as it is the sum of sorted vectors.
>
> **I found Remark 1 both interesting and important, thus should be elaborated more in the paper.**
>
> Thank you for the suggestion and acknowledging the significance of this finding. As pointed out in the response to all reviewers,
> >This is an important finding for enabling usage of INB in the federated setting: INB backpropagates through $\boldsymbol{y}$ when computing $\nabla_{\boldsymbol{\theta}} \boldsymbol{d}_m$, which requires keeping track of how $\boldsymbol{y}$ is computed from each $\boldsymbol{x}_m$. This could lead to both communication and privacy concerns. Instead, we show that treating $\boldsymbol{y}$ as a constant does not lead to any change in the gradient computation, which alleviates these concerns.
> >
> One important point for Remark 1 that can be further elaborated is our use of $c(x, y)=||x-y||_2^2$ as transportation cost when defining Wasserstein distance, which is a common choice. Based on this, While the gradients computed at each client/domain are biased because the barycenter is treated as a constant, the sum of the gradients over all clients/domains is actually unbiased because the individual biases cancel out as we prove as we show in the formal proof of Remark 1 in Appendix B. We have added this discussion on the benefit and intuition of Remark 1 in Section 2.2 in the main paper.
>
> Please let us know if you have any further comments. We look forward to hearing from you again soon.

---

> ### Author Response · Authors · 2022-11-16
> **Follow-up**
>
> Hi Reviewer uckp, thank you for your helpful feedback. Have we adequately addressed your questions and concerns?

---

### Author Response · Authors · 2022-11-09
**Response to all**

We thank all the reviewers for their helpful comments and positive impressions of our work. Here, we wish to clarify the main novelty of our work in response to a few of the comments made and we have slightly modified the contribution in Section 1 accordingly (highlighted in blue). Then, we respond point-by-point to each reviewer below.

Our central contribution is **developing methods that address the novel problem of** ***FL translation*** that has not been considered in the literature before. Domain translation in FL is a very difficult task, particularly for the realistic use case where each client represents a domain, as we demonstrate empirically in the paper. **Typical translation models cannot train under communication and privacy constraints.** This novel problem setting motivates our choice of algorithm (i.e., INB over standard translation models), our theoretical developments (e.g., showing that the Fed algorithm is equivalent to the non-Fed algorithm), and our engineering innovations (e.g., variable-bin-width histograms, sorting before transmission for privacy, and pseudo-empirical barycenter samples for communication and privacy).

More specifically, first, we prove that federated multi-max-K-SW is equivalent to the original multi-max-K-SW. This is an **important finding for enabling usage of INB in the federated setting**: INB backpropagates through $\boldsymbol{y}$ when computing $\nabla_{\boldsymbol{\theta}} \boldsymbol{d}_m$, which requires keeping track of how $\boldsymbol{y}$ is computed from each $\boldsymbol{x}_m$. This could lead to both communication and privacy concerns. Instead, we show that treating $\boldsymbol{y}$ as a constant does not lead to any change in the gradient computation, which alleviates these concerns. Second, we propose to use **VW histograms** instead of directly sending samples over the network. This significantly improves privacy and resource efficiency---again, an innovation that is novel compared to INB and is motivated by FL concerns. We empirically show that the performance does not decrease with this FL-motivated innovation. Third, we **empirically demonstrate that our FedINB approach performs significantly better** than FL versions of standard translation models under settings with limited client-server communication capability, which is of critical practical importance in FL. Fourth, we **empirically demonstrate the feasibility of FedINB for federated domain generalization**, which is a challenging practical problem with few existing solutions. We also show that our federated DG method provides substantial improvements in robustness to an increasing synchronization period, allowing reductions in communication overhead, which is again important in FL settings.

Please let us know if you have any further comments. We look forward to hearing from you again soon.

---

### Author Response · Authors · 2022-11-29
**Follow-up to all reviewers**

Hi all reviewers, we want to update here that in response to Reviewer CNpn's follow-up question, we add an extra experiment in Appendix F.6 to check if the convergence of the inner problem (i.e., multi-max-K-SW) is fundamentally changed by our VW histograms and AE innovations. From our investigations, VW histograms do not seem to hurt the convergence of multi-max-K-SW and autoencoders lead to a faster convergence of multi-max-K-SW. More details can be found in the revised manuscript in Appendix F.6.

Additionally, as we are approaching the end of Discussion Stage 2, we would like to kindly ask whether we have adequately addressed your comments?

---

### Decision · Program_Chairs · 2023-01-20

**Decision:**

Accept: poster

**Justification For Why Not Higher Score:**

While reviewers are consistently positive and agree, we don't have high enough confidence for spotlight level

**Justification For Why Not Lower Score:**

Reviewers were consistently positive and agree

**Metareview: Summary, Strengths And Weaknesses:**

The paper studies an iterative domain translation model, adapted to federated learning. It aims to allow transfer between stronger distribution shifts, such as conditional shifts.
To do so, it relies on a federated version of the Iterative Naive Barycenter (INB) method, which maps each domain distribution to a shared latent distribution.

Reviewers were overall positive yet not very high confidence.

We hope the authors will incorporate the several points mentioned by the reviewers in the final version.

**Note From Pc:**

if the above contains the word "oral" or "spotlight" please see: "oral" presentation means -> notable-top-5% and "spotlight" means -> notable-top-25%. As stated in our emails, we are disassociating presentation type from AC recommendations